# ODYSSEY: EMPOWERING MINECRAFT AGENTS WITH OPEN-WORLD SKILLS

## ABSTRACT

Recent studies have delved into constructing generalist agents for open-world environments like Minecraft. Despite the encouraging results, existing efforts mainly focus on solving basic programmatic tasks, *e.g.*, material collection and tool-crafting following the Minecraft tech-tree, treating the `ObtainDiamond` task as the ultimate goal. This limitation stems from the narrowly defined set of actions available to agents, requiring them to learn effective long-horizon strategies from scratch. Consequently, discovering diverse gameplay opportunities in the open world becomes challenging. In this work, we introduce ODYSSEY, a new framework that empowers Large Language Model (LLM)-based agents with open-world skills to explore the vast Minecraft world. ODYSSEY comprises three key parts: (1) An interactive agent with an *open-world skill library* that consists of 40 primitive skills and 183 compositional skills. (2) A fine-tuned LLaMA-3 model trained on a *large question-answering dataset* with 390k+ instruction entries derived from the Minecraft Wiki. (3) A *new agent capability benchmark* includes the long-term planning task, the dynamic-immediate planning task, and the autonomous exploration task. Extensive experiments demonstrate that the proposed ODYSSEY framework can effectively evaluate different capabilities of LLM-based agents. All datasets, model weights, and code are publicly available to motivate future research on more advanced autonomous agent solutions.

## 1 INTRODUCTION

Developing autonomous agents capable of performing open-world tasks represents a significant milestone towards achieving artificial general intelligence (Savva et al., 2019; Reed et al., 2022; Driess et al., 2023). These open-world tasks necessitate that agents interact with complex and dynamic environments, make decisions based on incomplete information, and adapt to unexpected events. Early reinforcement learning agents (Tessler et al., 2017; Oh et al., 2017; Guss et al., 2019) have demonstrated limited knowledge in such open-world setting. Furthermore, these agents often struggle with long-term planning, which is crucial for the fulfillment of intricate goals. Recent breakthrough of Large Language Models (LLMs) (Hu et al., 2021; Achiam et al., 2023; Touvron et al., 2023) have shown the potential to revolutionize various fields such as healthcare (Zhang et al., 2023b; Yang et al., 2024b), robotics (Huang et al., 2022; Ahn et al., 2022; Singh et al., 2023), and web services (Nakano et al., 2021; Deng et al., 2023; Iong et al., 2024), attributed to its capability on endowing agents with expansive knowledge and sophisticated planning akin to human reasoning (Wei et al., 2022a; Wang et al., 2024a; Liang et al., 2023). However, the development of LLMs in open-world tasks remains challenging due to the need for well-defined environments and measurable benchmarks (Zhu et al., 2023; Wang et al., 2023a; Qin et al., 2023).

The popular Minecraft game features a vast and diverse world with various biomes, terrains, and resources, making it an ideal testbed for evaluating the capabilities of autonomous agents in the open-world setting (Guss et al., 2019). To facilitate the development of generalist agents in this setting, MineRL (Guss et al., 2019) and MineDojo (Fan et al., 2022) introduced simulation benchmarks built upon the sandbox Minecraft environment. The seminal work, Voyager (Wang et al., 2023a), proposed an LLM-based agent to drive exploration in Minecraft. Subsequently, there has been a surge of efforts to leverage the superior performance of LLMs to extend the capabilities of such Minecraft agents (Zhu et al., 2023; Wang et al., 2023b; Zhou et al., 2024a; Wang et al., 2023c; Qin et al., 2023). Despite recent advancements, existing works mainly focus on solving basic pro-

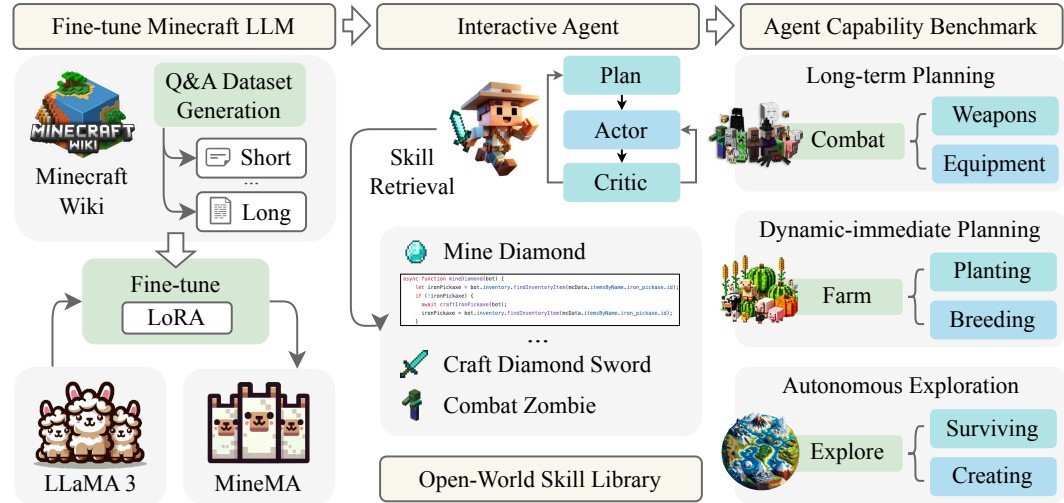

Figure 1: An overview of the proposed ODYSSEY framework. Odyssey consists of three key components: (1) a fine-tuned LLaMA-3 model trained on a large-scale question-answering dataset; (2) an interactive agent equipped with an extensive open-world skill library; (3) a novel agent capability benchmark encompassing a variety of tasks.

grammatic tasks, often considering the `ObtainDiamond` task as the ultimate challenge. Basic programmatic tasks refer to those constrained by the explicit dependencies following the Minecraft tech-tree, such as collecting materials and crafting tools. Such tasks inherently only assess the ability of LLMs to prioritize crafting steps within a limited task space, rather than their potential for complicated and diverse solutions. This limitation arises from the narrowly defined set of actions available to agents (*e.g.*, mouse and keyboard), which necessitates learning skills from scratch. Since Minecraft is fundamentally resource-based, an agent must first learn to collect adequate resources and tools to engage in creative play, which limits the exploration of diverse gameplay options. Moreover, methods like Voyager (Wang et al., 2023a) heavily rely on the powerful GPT-4 for high-quality solutions, imposing a substantial cost burden on researchers who prefer open-source models.

In this work, we introduce ODYSSEY[1], a novel framework that equips LLM-based agents with advanced open-world skills, enabling efficient interaction and exploration within the Minecraft environment. ODYSSEY allows agents to move beyond basic programmatic tasks and focus more on complex open-world challenges. As shown in Fig. 1, ODYSSEY comprises three key contributions:

1. We develop an LLM-based interactive agent with an *open-world skill library*, encompassing 40 primitive skills that serve as underlying interfaces and 183 compositional skills tailored for complex and diverse tasks in an open-world setting. A recursive method improves skill execution by checking prerequisites. The ODYSSEY agent consists of a planner for goal decomposition, an actor for skill retrieval and subgoal execution, and a critic for feedback and strategy refinement.

2. We fine-tune the LLaMA-3 model (Touvron et al., 2023) for Minecraft agents using a *comprehensive question-answering dataset*. This involves generating a large-scale training dataset with 390k+ instruction entries from Minecraft Wikis, fine-tuning various sizes of the LLaMA-3 models using LoRA (Hu et al., 2021), and evaluating them with a custom multiple-choice dataset.

3. We introduce a *new agent capability benchmark* to evaluate different aspects of agent performance in Minecraft, including the long-term planning task, the dynamic-immediate planning task, and the autonomous exploration task. Extensive experiments demonstrate that the proposed ODYSSEY framework provides a robust measure of agent effectiveness, showcasing the practical advantages of our framework using the open-source models.

It is worth noting that our focus is *not to design a new LLM-based agent architecture*. Instead, this work aims to provide a comprehensive framework for *developing and evaluating autonomous agents in open-world environments*, enabling them to explore the vast and diverse Minecraft world.

---

[1]The Odyssey is a great ancient Greek epic poem attributed to Homer, which is now often used metaphorically to describe *a long adventurous journey* (Oxford English Dictionary).

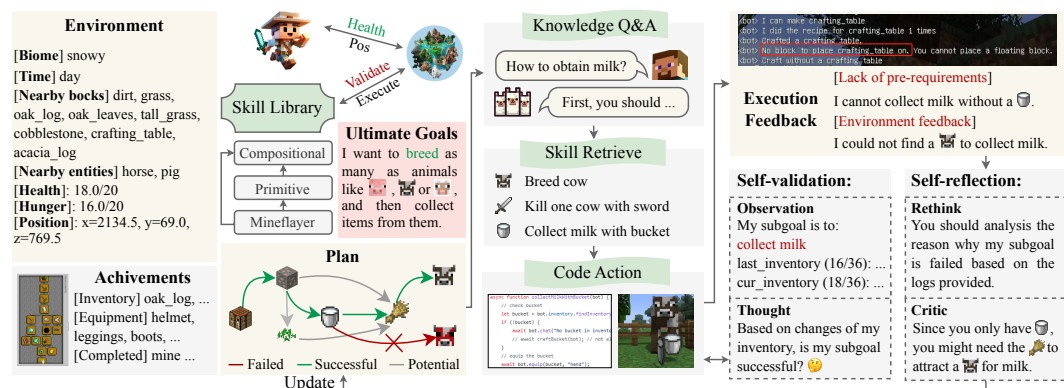

Figure 2: An illustrative diagram of the interactive agent following a planner-actor-critic architecture based on the open-world skill library. The LLM Planner decomposes ultimate goals into specific subgoals, while the LLM Actor then sequentially executes code actions for each subgoal using the skill library. The LLM Critic evaluates these actions through self-validation and reflection, enabling the agent to update its plan based on execution feedback.

We have open-sourced all parts of ODYSSEY and will continuously update the repository. We hope this will enable other researchers to build upon our work, fostering further innovation and progress in the development of autonomous agents.

# 2 OPEN-WORLD SKILL-BASED INTERACTIVE AGENT

ODYSSEY develops an LLM-based interactive agent with an open-world skill library, aiming to enhance the efficiency and adaptability of agents in complex Minecraft environments. The skill library comprises 40 primitive skills and 183 compositional skills, while the LLM-based agent employs a planner-actor-critic architecture to facilitate task decomposition, skill execution, and performance feedback. The architecture of the interactive agent is depicted in Fig. 2. Full skill and prompt details used in the LLM-based interactive agent are given in Appendix C.

## 2.1 OPEN-WORLD SKILL LIBRARY

**Primitive skills** encompass a series of underlying interfaces on top of Mineflayer JavaScript APIs (PrismarineJS, 2023), divided into two main categories: 32 operational skills and 8 spatial skills. This suite of skills exceeds the 18 primitive skills (all are operational skills) delineated in Voyager (Wang et al., 2023a). Operational skills serve as foundational interfaces with parameterized input, such as `mine(·)` for material collection and `craft(·)` for tool crafting. Additionally, we pioneer 8 spatial skills that Voyager (Wang et al., 2023a) lacks, allowing for environmental interactions based on the agent coordinates. Given that our work is conducted within a text-based Minecraft environment (Wang et al., 2023a; Fan et al., 2022), spatial skills are crucial for handling tasks that require precise positioning and orientation, especially in the absence of visual input.

**Compositional skills** encapsulate primitive skills into higher-level ones, functioning to address a variety of basic programmatic tasks, such as `mineDiamond` and `craftIronPickaxe`. ODYSSEY classifies 183 compositional skills into types like `mineX`, `craftX`, `plantX`, `breedX`, `cookX`, *etc*. We use a recursive method to construct the skill library, simplifying complex task decomposition by ensuring prerequisites are met before skill execution. Taking `mineDiamond` as an example, if the agent lacks an iron pickaxe, it will recursively execute `craftIronPickaxe`. This indicates that our program internally manages the construction and execution order of skills through its recursive method, thereby avoiding the need for the agent to engage in additional planning.

To facilitate efficient retrieval of skills in the skill library, we first generate a description for each skill by calling the LLM and using the complete program code as a prompt. We then employ Sentence Transformer (Reimers & Gurevych, 2019) to encode the skill description. This method transforms text information into vector representations, facilitating semantic retrieval and enabling the agent to find the most relevant skill description based on the context provided.

## 2.2 PLANNER-ACTOR-CRITIC ARCHITECTURE

**LLM Planner.** The LLM Planner is responsible for developing a comprehensive plan, facilitating efficient exploration through long-term goal decomposition. The LLM Planner breaks down high-level goals into specific low-level subgoals, each corresponding to a particular skill outlined in Sec. 2.1. By addressing each subgoal in the plan, the ultimate goal can be progressively achieved. The input prompt to the planner consists of several components: **(1) Ultimate goals and behavioral constraints.** For example, "My ultimate goal is to ... Propose the current task only when you ensure that you have all the necessary dependent items in inventory". **(2) States of the agent.** This reflects the interaction between the agent and environment, such as hunger and health values, position and nearby entities, *etc*. **(3) Achievements of the agent.** This includes the current inventory and unlocked equipment, as well as previously successful and failed tasks.

**LLM Actor.** In the execution phase, the LLM actor is invoked to sequentially execute the subgoals generated by the LLM planner within the Minecraft environment. This process utilizes the open-world skill library to achieve these subgoals. The mapping from high-level subgoals to executable skill code is accomplished through query context encoding and skill similarity retrieval. This process includes: **(1) Query context.** The text-based subgoals generated by the LLM planner are encoded by Sentence Transformer (Reimers & Gurevych, 2019) to vector representations as the query context. **(2) Similarity matching.** The vector similarity between the query context and the skill descriptions in the skill library is computed to determine semantic closeness. **(3) Skill selection.** The top-5 relevant skills with the highest scores are identified, and the actor agent selects the most appropriate code for execution within the environment based on their descriptions.

**LLM Critic.** During action execution, it is critical for an agent to document its experiences, especially noting successful outcomes and failure points. This is crucial in open-world planning to establish a feedback-informed system, which corrects initial plan discrepancies that can cause execution errors. For instance, achieving the animal breeding goal requires prerequisite crops for feed. The LLM critic can assess action effectiveness by comparing expected and actual outcomes, providing insights for refining future strategies. We categorize feedback into three types: **(1) Execution feedback.** This captures the progress of skill execution. For example, "No hoe in inventory. Craft a hoe first!" not only highlights the reason for failure in hoeing farmland but also provides a guideline to address this problem. **(2) Self-validation.** By presenting inventory changes post-action to the LLM critic, we empower it to validate whether the skill has achieved its subgoal, eliminating the need for manual checks. **(3) Self-reflection.** Simply confirming the completion of a subgoal is often inadequate for correcting planning errors. The LLM critic also serves as an analyst, deducing the cause of task failure by evaluating the current state of the agent and its environment. It then offers a critique, suggesting a more efficient strategy for task completion.

## 3 FINE-TUNE MINECRAFT LLM

To improve agent performance in Minecraft, we fine-tune the LLaMA-3 model (Touvron et al., 2023) using a large-scale Question-Answering (Q&A) dataset with 390k+ instruction entries sourced from the Minecraft Wiki. ODYSSEY presents an effective procedure for converting a foundation model into a domain-specific model, which involves dataset generation, model fine-tuning, and model evaluation. The detailed descriptions can be found in Appendix D.

**Dataset Generation.** We develop a GPT-assisted method to generate an instruction dataset for Minecraft. First, we crawl relevant content from the Minecraft Wiki, excluding non-essential sections like history. The collected data is then categorized and separated into different files based on their content type. Then we use GPT-3.5-Turbo (OpenAI, 2023) with different customized prompts to automatically generate diverse Q&A pairs. Note that both the questions and answers were generated by GPT. These Q&A pairs are categorized into four types based on the nature of the answers: short, normal, long, and boolean, yielding 390k+ entries. In contrast, the Wiki dataset released by MineDojo (Fan et al., 2022) only collects Minecraft Wiki pages, without refining the content and generating Q&A pairs for model training. STEVE (Zhao et al., 2023) introduces a non-public dataset with 20k+ Q&A pairs, which is smaller than our dataset in terms of scale and diversity.

**Model Fine-tuning.** We employ LoRA (Hu et al., 2021) for model fine-tuning, which is a parameter-efficient training technique. LoRA introduces small, trainable low-rank matrices to adapt a pre-

trained neural network, enabling targeted updates without the need to retrain the entire model. Using LoRA, we fine-tune the LLaMA-3-8B-Instruct and LLaMA-3-70B-Instruct models with our Minecraft dataset, resulting in the new models termed MineMA-8B and MineMA-70B, respectively.

**Model Evaluation.** In Minecraft, questions are often open-ended and can yield diverse answers; therefore, conventional evaluation metrics (Papineni et al., 2002; Lin, 2004) may fall short. Meanwhile, common benchmarks (Wang et al., 2018; 2019; Hendrycks et al., 2021) are not suitable for assessing the capabilities of expert models. Thus, we employed GPT-4 (Achiam et al., 2023) to generate two Multiple-Choice Question (MCQ) datasets based on different themes and keywords related to Minecraft. These datasets can quantitatively evaluate the domain-specific expertise of models.

# 4 AGENT CAPABILITY BENCHMARK

ODYSSEY presents a new benchmark for evaluating agent capabilities within Minecraft, offering three task types: long-term planning, dynamic-immediate planning, and autonomous exploration. It is notable that these tasks cannot be solved by any single skill but demand a sophisticated combination of multiple skills. These tasks are set in various Minecraft scenarios, with different tasks in the same scenario testing different agent capabilities. For example, in the cooking scenario, long-term planning requires formulating a complete plan to locate and hunt a specific animal, whereas dynamic-immediate planning involves selecting which nearby animal to cook based on the immediate environment. Our benchmark provides a standardized framework for evaluating agents, where the agent capability requirements for different tasks are shown in Table 1. Please refer to Appendix E for more details.

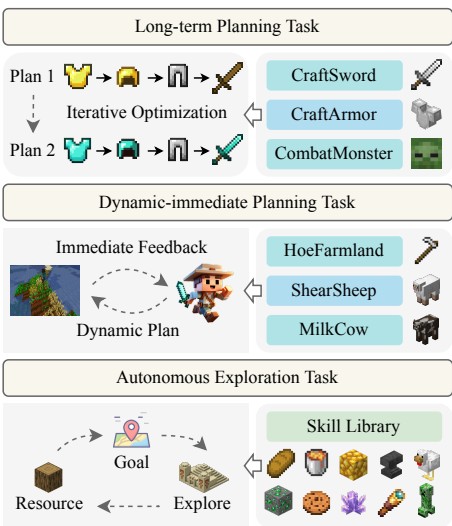

Figure 3: Agent capability benchmark.

**Long-term Planning Task.** We design a suite of combat scenarios to assess the long-term planning capability of agents, requiring them to craft appropriate weapons and equipment to defeat various monsters. These combat scenarios can be divided into single-type and multi-type monster scenarios. For the single-type scenarios, we choose various unique monsters, each with its own attack styles, movement patterns, and hostility levels. For the multi-type scenarios, we focus on typical monster groupings encountered in the game. Agents must generate a comprehensive long-term plan, detailing the sequence of crafting the necessary weapons and equipment for the assigned combat task. Performance is measured by remaining health and time consumed during combat. After each battle, agents can iteratively optimize their plan, learning from previous outcomes to improve performance in subsequent rounds. To extend the scope of the long-term planning task beyond combat, we also adopt animal husbandry and cooking scenarios, where agents are required to formulate detailed plans for completing tasks related to specific animals.

**Dynamic-immediate Planning Task.** The dynamic-immediate planning task requires agents to dynamically generate and execute plans based on immediate environmental feedback. Thus, we design a suite of farming scenarios, where agents engage in activities like planting, cooking, and animal husbandry. Although some scenarios are similar to the long-term planning task, the dynamic-immediate planning task emphasizes reacting to real-time feedback like available resources and nearby animals. Performance is evaluated through task completion time and success rates.

**Autonomous Exploration Task.** To test the exploratory capability of agents within open-world settings, we design an autonomous exploration task in Minecraft. In this task, agents are required to determine their subsequent objectives and execute the appropriate skills based on the game context. The exploration task involves discovering and utilizing resources, while adapting to unexpected events such as encounters with hostile monsters. Agents must adapt to these challenges by developing strategies for resource management and task prioritization. The performance metrics include the number of distinct items obtained, the total items crafted, the recipes and advancements (R&A) unlocked, and the distance traveled.

Table 1: Specific agent capability requirements for different benchmark tasks, including Goal-based Planning (GBP), Feedback-based Planning (FBP), Exploratory Planning (EP), Task Decomposition (TD), Resource Management (RM), Skill Retrieval (SR), Self-Reflection (Self-R), and Self-Validation (Self-V). Please refer to Appendix E.4 for detailed descriptions of each capability.

| Task | GBP | FBP | EP | TD | RM | SR | Self-R | Self-V |
|------|-----|-----|-----|-----|-----|-----|--------|--------|
| Single-Round Long-Term Planning Task | ✓ | × | × | ✓ | × | ✓ | ✓ | ✓ |
| Multi-Round Long-Term Planning Task | ✓ | ✓ | × | ✓ | × | ✓ | ✓ | ✓ |
| Dynamic-Immediate Planning Task | ✓ | ✓ | × | × | ✓ | ✓ | ✓ | ✓ |
| Autonomous Exploration Task | × | ✓ | ✓ | × | ✓ | ✓ | ✓ | ✓ |

Table 2: Average execution time and success rate (SR) on 5 basic programmatic tasks in Minecraft.

| Task | Time (min) | SR in 2min | SR in 5min | SR in 10min | SR in 15min |
|------|-----------|-----------|-----------|------------|------------|
| 📦 Crafting Table | 0.59 ± 0.79 | 95.8% | 99.2% | 100.0% | 100.0% |
| ➶ Wooden Tool | 0.95 ± 0.80 | 92.5% | 99.2% | 100.0% | 100.0% |
| ➶ Stone Tool | 1.48 ± 0.96 | 85.0% | 97.5% | 100.0% | 100.0% |
| ➶ Iron Tool | 4.43 ± 1.48 | 0.0% | 76.7% | 100.0% | 100.0% |
| 💠 Obtain Diamond | 6.48 ± 2.02 | 0.0% | 21.7% | 92.5% | 100.0% |

## 5 EXPERIMENTS

To demonstrate the effectiveness of the proposed ODYSSEY framework, we conduct experiments on basic programmatic tasks and the agent capability benchmark. Our simulation environment is built on top of Voyager (Wang et al., 2023a), providing a text-based interface for agents to interact with Minecraft. We only use GPT-3.5 and GPT-4 for initial data generation, but all experiments are conducted with the open-source LLaMA-3 model, significantly reducing costs compared to GPT-4-based skill generation methods (Wang et al., 2023a;b). Notably, we do not employ GPT-4 in Voyager due to the high cost, which we estimate would be in the thousands of dollars per experiment. Instead, we reproduce Voyager using GPT-4o-mini and GPT-3.5 for comparison. More details are provided in Appendix F. We aim to answer the following questions: (1) Can the open-world skill library improve the efficiency of agents in Minecraft? (Sec. 5.1). (2) How well do agents with different LLMs perform on the agent capability benchmark tasks? (Sec. 5.2). (3) What is the contribution of different components of the ODYSSEY agent to its overall performance? (Sec. 5.3).

### 5.1 OPEN-WORLD SKILL LIBRARY

To demonstrate the superior capability of our open-world skill library in Minecraft, we first tested it on 5 basic programmatic tasks from previous studies (Zhu et al., 2023). We conducted 120 repeated experiments on each task and recorded the average completion time for each task as well as the success rates at different time points. The results in Table 2 demonstrate that our open-world skill library efficiently handles basic programmatic tasks. Simple tasks achieve near-perfect success within five minutes. Even for difficult tasks like obtaining a diamond, success rates rise from 21.7% at five minutes to 92.5% at ten minutes, highlighting the effectiveness of the skill library.

### 5.2 AGENT CAPABILITY BENCHMARK

We evaluate the LLM-based agent on the long-term planning task, the dynamic-immediate planning task, and the autonomous exploration task from the ODYSSEY benchmark. These tasks cover a variety of complex gaming scenarios and require diverse solutions.

#### 5.2.1 LONG-TERM PLANNING TASK

The long-term planning task assesses the agent capability to directly formulate and execute comprehensive plans over extended periods. For example, in the combat scenarios, the agent is required to plan a list of weapons and equipment to craft based on the strength of different monsters, with the

Table 3: Performance comparison of different models on the single-round long-term planning task. "Health" refers to the remaining health points. "# LLM iters" is the number of LLM iterations (calling LLM) required to complete the task. "Time (min)" refers to the minutes spent in both gathering materials and crafting equipment to defeat different monsters. All evaluation metrics are calculated only for successful tasks. $\pm$ corresponds to one standard deviation of the average evaluation over successful tasks. **Bold** and *italics* mean the best and the second-best results. "-" indicates that health is not a relevant metric in the scenarios. "N/A" indicates that all tasks fail.

| Task | Model | Success Rate | Health | Time (min) | # LLM Iters |
|---|---|---|---|---|---|
| 1 zombie | Voyager | **3 / 3** | **20.0 ± 0.0** | 9.9 ± 6.0 | 67.3 ± 41.7 |
| | LLaMA-3-8B | 4 / 8 | **20.0 ± 0.0** | **8.3 ± 4.2** | **6.1 ± 4.1** |
| | MineMA-8B | **8 / 8** | 19.4 ± 2.3 | *8.8 ± 5.4* | *10.0 ± 5.8* |
| 1 spider | Voyager | **3 / 3** | 10.8 ± 8.0 | *9.4 ± 8.8* | 19.0 ± 1.4 |
| | LLaMA-3-8B | 4 / 8 | **19.4 ± 1.0** | 12.1 ± 3.8 | **8.4 ± 3.5** |
| | MineMA-8B | **8 / 8** | *19.3 ± 1.6* | **8.3 ± 6.7** | *15.2 ± 6.0* |
| 1 skeleton | Voyager | *2 / 3* | *16.5 ± 0.0* | **7.4 ± 2.9** | 46.0 ± 32.0 |
| | LLaMA-3-8B | 4 / 8 | **17.6 ± 2.7** | *8.1 ± 3.5* | **8.9 ± 3.7** |
| | MineMA-8B | **8 / 8** | 13.6 ± 5.9 | 8.6 ± 7.3 | *12.1 ± 7.0* |
| 1 zomb-ified piglin | Voyager | **3 / 3** | *19.0 ± 1.4* | 14.5 ± 4.7 | 50.3 ± 26.2 |
| | LLaMA-3-8B | 4 / 8 | **19.9 ± 0.4** | *9.2 ± 3.9* | **10.0 ± 4.2** |
| | MineMA-8B | **8 / 8** | 18.7 ± 1.9 | **8.5 ± 6.1** | *11.7 ± 6.2* |
| 1 ender-man | Voyager | *2 / 3* | 11.0 ± 9.0 | 22.8 ± 1.7 | 28.0 ± 4.0 |
| | LLaMA-3-8B | 2 / 8 | *15.1 ± 7.3* | *13.0 ± 3.0* | **6.8 ± 1.9** |
| | MineMA-8B | *4 / 8* | **19.8 ± 0.5** | **10.4 ± 6.3** | *12.5 ± 5.4* |
| 1 zombie villager | Voyager | 2 / 3 | **20.0 ± 0.0** | *12.6 ± 2.0* | 50.0 ± 3.0 |
| | LLaMA-3-8B | 7 / 8 | 19.6 ± 1.1 | 12.7 ± 5.3 | **11.0 ± 5.3** |
| | MineMA-8B | **8 / 8** | **20.0 ± 0.0** | **9.0 ± 3.6** | *12.8 ± 6.1* |
| 1 cave spider | Voyager | 2 / 3 | 16.5 ± 3.5 | *10.0 ± 1.8* | 79.2 ± 29.0 |
| | LLaMA-3-8B | *6 / 8* | *19.5 ± 1.2* | 12.0 ± 6.3 | *19.5 ± 1.2* |
| | MineMA-8B | **7 / 8** | **20.0 ± 0.0** | **3.6 ± 2.6** | **8.6 ± 8.8** |
| 1 wither skeleton | Voyager | 1 / 3 | **20.0 ± 0.0** | 20.9 ± 0.0 | 100.0 ± 0.0 |
| | LLaMA-3-8B | *6 / 8* | 13.2 ± 6.0 | *11.7 ± 3.7* | **12.3 ± 2.7** |
| | MineMA-8B | **7 / 8** | *17.3 ± 3.7* | **11.0 ± 6.8** | *12.6 ± 6.9* |
| 1 zombie, 1 spider | Voyager | *1 / 3* | *17.5 ± 0.0* | **5.9 ± 0.0** | 21.0 ± 0.0 |
| | LLaMA-3-8B | 1 / 8 | **20.0 ± 0.0** | *8.5 ± 0.0* | **6.0 ± 0.0** |
| | MineMA-8B | **5 / 8** | 16.4 ± 4.1 | 10.6 ± 6.7 | *12.0 ± 4.9* |
| 1 zombie, 1 skeleton | Voyager | **2 / 3** | **19.0 ± 1.0** | 15.0 ± 8.6 | 40.5 ± 20.5 |
| | LLaMA-3-8B | 1 / 8 | 0.2 ± 0.0 | **13.5 ± 0.0** | **9.0 ± 0.0** |
| | MineMA-8B | *3 / 8* | *12.8 ± 2.8* | *14.0 ± 1.9* | *10.3 ± 2.8* |
| 3 zombies | Voyager | **2 / 3** | **7.8 ± 4.2** | **8.2 ± 0.4** | 61.0 ± 29.0 |
| | LLaMA-3-8B | *1 / 8* | *3.7 ± 0.0* | 14.3 ± 0.0 | **8.0 ± 0.0** |
| | MineMA-8B | *1 / 8* | 5.2 ± 0.0 | *11.1 ± 0.0* | *14.0 ± 0.0* |
| cook meat | Voyager | 0 / 3 | - | N/A | N/A |
| | LLaMA-3-8B | *1 / 8* | - | **20.3 ± 0.0** | **19.0 ± 0.0** |
| | Minema-8B | **2 / 8** | - | *21.4 ± 1.2* | *30.0 ± 10.0* |
| animal husbandry | Voyager | *1 / 3* | - | 19.0 ± 0.0 | **12.0 ± 0.0** |
| | LLaMA-3-8B | 2 / 8 | - | **15.3 ± 7.6** | 31.0 ± 4.0 |
| | Minema-8B | **3 / 8** | - | *16.8 ± 7.8* | *26.7 ± 16.2* |

goal of defeating the monster in as short a time as possible. We compared the performance of our agent with both the fine-tuned MineMA-8B and the original LLaMA-3-8B models, and also the performance of Voyager (Wang et al., 2023a) with GPT-4o-mini across these tasks. Moreover, we also evaluate the performance of single-round and multi-round planning. The single-round test results in

Tab. 3 demonstrate that the fine-tuned MineMA-8B model surpasses the original LLaMA-3-8B model in terms of success rate and time efficiency, albeit at the cost of more LLM iterations. Moreover, our agent with the MineMA-8B model can outperform Voyager with GPT-4o-mini in most scenarios, indicating the effectiveness of our fine-tuning strategy. The multi-round test results in Fig. 4 demonstrate that the multi-round planning strategy significantly improves the time efficiency of the agent. This improvement suggests that the agent is capable of iteratively refining its plans based on the outcomes of previous encounters, thereby boosting its performance in subsequent rounds.

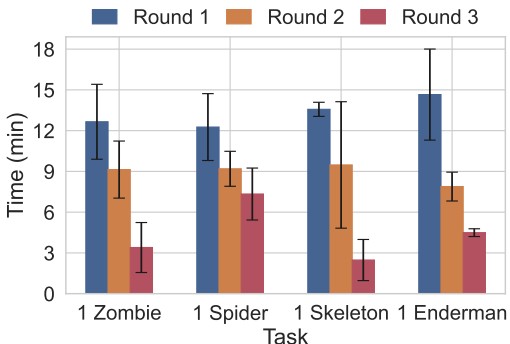

Figure 4: Performance on the multi-round long-term planning task. Note that all presented data are from successful tasks.

### 5.2.2 DYNAMIC-IMMEDIATE PLANNING TASK

For the dynamic-immediate planning task, the agent is required to dynamically generate and execute plans based on immediate environmental feedback. We compared our MineMA model with different open-sourced LLMs, including GPT-4o, Qwen2-7B (Yang et al., 2024a) and Baichuan2-7B (Yang et al., 2023). Moreover, we evaluate the performance of the MineMA-8B and the MineMA-70B model to investigate the impact of model size on task performance. As shown in Tab. 4, the MineMA-8B model outperforms the Baichuan2-7B and Qwen2-7B models in terms of success rate and time efficiency. Moreover, the MineMA-70B model shows superior performance compared with the MineMA-8B model. Across all open-sourced LLMs, MineMA-70B demonstrates higher success rates and generally lower average execution times and LLM iterations. Additionally, our MineMA can achieve performance similar to that of GPT-4o.

### 5.2.3 AUTONOMOUS EXPLORATION TASK

In the autonomous exploration task, the agent is required to explore the Minecraft world freely without any specific goals. We compare our agent with different Minecraft-based agent methods (Voyager (Wang et al., 2023a) and DEPS (Wang et al., 2023b)) and different LLM-based agent techniques (ReAct (Yao et al., 2023) and AutoGPT (Significant-Gravitas, 2023)) on this task. Note that we reproduced different LLM-based agent techniques following the same settings as in Voyager (Wang et al., 2023a). As shown in Fig. 5, our agent with the MineMA-8B model can achieve superior performance compared with all baselines, indicating that the agent can autonomously explore the Minecraft world without specific goals. It is notable that our agent with the MineMA-8B model can outperform Voyager (Wang et al., 2023a) with GPT-4o-mini or GPT-3.5.

### 5.3 ABLATION STUDY

We conduct ablation studies on two core components of the ODYSSEY agent, including the LLM planner and the open-world skill library. The results are shown in Fig. 5. In the autonomous exploration task, the LLM planner is responsible for generating a comprehensive plan based on the open-world skill library. The ablation study demonstrates that the planner is indispensable for the agent to effectively navigate the complex Minecraft environment. Additionally, our experimental results indicate that the absence of the open-world skill library significantly degrades performance. Without the open-world skill library, the 8B LLM model alone is largely incapable of generating executable codes for the agent. This underscores the critical role of the open-world skill library in enabling the agent to perform complex tasks within the open-world setting of Minecraft.

## 6 RELATED WORKS

**Minecraft agents** have been widely studied in recent years to test the capabilities of autonomous agents in open-world environments. Previous works focused on training Minecraft agents with reinforcement learning (Tessler et al., 2017; Oh et al., 2017; Lin et al., 2022; Mao et al., 2022; Hafner

Table 4: Performance comparison of different models on the dynamic-immediate planning task. All evaluation metrics are calculated only for successful tasks. **Bold** and *italics* mean the best and the second-best results of all open-sourced LLMs (excluding GPT-4o). "N/A" indicates that all tasks fail. Please refer to Appendix F.4 for easier visual inspection.

| Task | Model | Success Rate | Time (min) | # LLM Iters |
|---|---|---|---|---|
| Collect Seeds | GPT-4o | 5 / 5 | 1.2 ± 0.5 | 1.0 ± 0.0 |
| | Baichuan2-7B | 2 / 5 | 1.8 ± 1.4 | 3.0 ± 2.8 |
| | Qwen2-7B | 2 / 5 | 3.8 ± 1.5 | 4.5 ± 0.7 |
| | MineMA-8B | **5 / 5** | **1.3 ± 1.4** | *1.4 ± 0.9* |
| | MineMA-70B | **5 / 5** | *1.4 ± 1.6* | **1.0 ± 0.0** |
| Hoe Farmland | GPT-4o | 5 / 5 | 3.9 ± 3.3 | 5.8 ± 4.7 |
| | Baichuan2-7B | 0 / 5 | N/A | N/A |
| | Qwen2-7B | *2 / 5* | *15.7 ± 16.2* | *19.5 ± 10.6* |
| | MineMA-8B | *2 / 5* | 17.2 ± 14.7 | 26.5 ± 9.2 |
| | MineMA-70B | **4 / 5** | **10.2 ± 6.7** | **11.8 ± 2.6** |
| Shear Sheep | GPT-4o | 5 / 5 | 4.7 ± 3.6 | 5.6 ± 6.5 |
| | Baichuan2-7B | 1 / 5 | 26.0 ± 0.0 | 30.0 ± 0.0 |
| | Qwen2-7B | *2 / 5* | *11.0 ± 2.8* | **10.8 ± 1.5** |
| | MineMA-8B | *2 / 5* | 15.7 ± 10.9 | 13.0 ± 9.9 |
| | MineMA-70B | **3 / 5** | **6.9 ± 7.8** | *11.0 ± 7.5* |
| Milk Cow | GPT-4o | 3 / 5 | 17.9 ± 8.3 | 20.3 ± 9.1 |
| | Baichuan2-7B | 0 / 5 | N/A | N/A |
| | Qwen2-7B | *1 / 5* | 26.1 ± 0.0 | 30.0 ± 0.0 |
| | MineMA-8B | *1 / 5* | **7.2 ± 0.0** | **7.0 ± 0.0** |
| | MineMA-70B | **2 / 5** | *8.6 ± 10.0* | *10.0 ± 11.3* |
| Cook Meat | GPT-4o | 3 / 5 | 5.5 ± 2.7 | 5.0 ± 4.2 |
| | Baichuan2-7B | 0 / 5 | N/A | N/A |
| | Qwen2-7B | 0 / 5 | N/A | N/A |
| | MineMA-8B | *1 / 5* | *25.6 ± 0.0* | *38.0 ± 0.0* |
| | MineMA-70B | **2 / 5** | **20.2 ± 8.5** | **24.0 ± 2.8** |
| Obtain Leather | GPT-4o | 5 / 5 | 14.8 ± 10.4 | 13.0 ± 8.2 |
| | Baichuan2-7B | 0 / 5 | N/A | N/A |
| | Qwen2-7B | 1 / 5 | *14.9 ± 0.0* | *16.0 ± 0.0* |
| | MineMA-8B | *4 / 5* | 15.0 ± 8.7 | 17.8 ± 15.2 |
| | MineMA-70B | **5 / 5** | **7.4 ± 7.8** | **8.8 ± 8.6** |
| Make Sugar | GPT-4o | 5 / 5 | 5.5 ± 3.6 | 7.0 ± 2.4 |
| | Baichuan2-7B | 2 / 5 | 16.2 ± 15.6 | 22.0 ± 18.4 |
| | Qwen2-7B | 2 / 5 | 15.4 ± 7.0 | 15.5 ± 9.2 |
| | MineMA-8B | **5 / 5** | **4.3 ± 1.9** | **7.0 ± 1.9** |
| | MineMA-70B | **5 / 5** | *4.3 ± 4.4* | *7.8 ± 4.0* |
| Collect Water | GPT-4o | 5 / 5 | 11.4 ± 1.6 | 27.3 ± 6.7 |
| | Baichuan2-7B | 0 / 5 | N/A | N/A |
| | Qwen2-7B | 1 / 5 | *10.0 ± 0.0* | 10.0 ± 0.0 |
| | MineMA-8B | *4 / 5* | 10.4 ± 3.0 | **8.8 ± 5.5** |
| | MineMA-70B | **5 / 5** | **9.3 ± 4.8** | *9.4 ± 3.7* |

et al., 2023) or imitation learning (Baker et al., 2022; Cai et al., 2023; Lifshitz et al., 2023), which are extensively used in the MineRL (Guss et al., 2019) competition to solve the `ObtainDiamond` task. With the rapid development of LLMs, numerous studies leverage LLMs to enhance agent capabilities (Zhang et al., 2023a; Zhu et al., 2023; Feng et al., 2023; Zhao et al., 2023; Wang et al., 2023a;b; Zheng et al., 2023; Zhou et al., 2024a; Li et al., 2024; Yu & Lu, 2024; Wang et al., 2024b; Cai et al., 2024). Among these, several works (Li et al., 2023; Yuan et al., 2023; Wang et al., 2023c; Qin et al., 2023; Ding et al., 2023) employ LLMs to guide skill learning in Minecraft, enabling

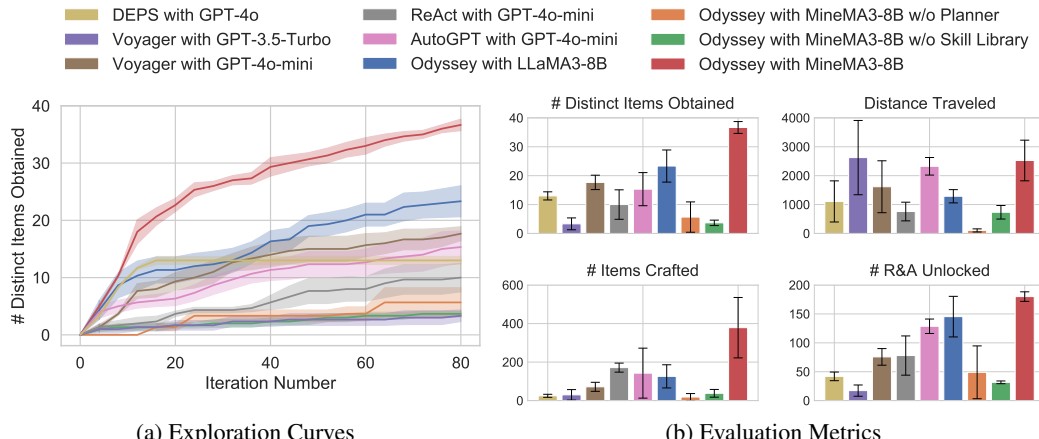

(a) Exploration Curves        (b) Evaluation Metrics

Figure 5: Performance comparison of different models on autonomous exploration tasks. To make the results in figures clearer for readers, we adopt a 50% confidence interval to plot the error region.

agents to act in a human-like way. However, these methods mainly focus on learning primitive skills from scratch, lacking a reusable skill library. Voyager (Wang et al., 2023a) builds a skill library by allowing the LLM to write its own skills. However, Voyager must rely on GPT-4 for high-quality skill generation, incurring substantial costs. This expense can be prohibitive for many researchers. In contrast, ODYSSEY provides an open-world skill library that agents can call upon, achieving performance comparable to Voyager with GPT-4, but using only 8B LLMs. This makes ODYSSEY significantly more accessible and cost-effective, enabling LLM-based agents to efficiently generate complex policies for broader exploration.

**Open-world environments** have gained considerable attention from research communities (Cao et al., 2020; Chevalier-Boisvert et al., 2018; Juliani et al., 2019; Shen et al., 2021; Srivastava et al., 2022; Du et al., 2023). Minecraft, with its diverse tasks and mature game mechanics, has emerged as an ideal test-bed for open-world tasks. Built on Minecraft, MineRL (Guss et al., 2019) implements a simulation environment for agent learning. MineDojo (Fan et al., 2022) further extends MineRL with thousands of diverse tasks. MCU (Lin et al., 2023) collects a variety of atom tasks, offering a method to generate infinite tasks by combining the atom tasks. However, existing benchmarks mainly focus on providing basic programmatic tasks to evaluate agents learned from scratch. Our ODYSSEY benchmark is built on top of the skill library, enabling the agents to bypass basic programmatic tasks and focus on complex open-world challenges.

## 7 CONCLUSION

This work proposes ODYSSEY to empower agents with open-world skills in the Minecraft environment. We introduce (1) an interactive agent endowed with an extensive open-world skill library comprising various primitive skills and compositional skills; (2) a fine-tuned LLaMA-3 model, trained on a large-scale question-answering dataset sourced from the Minecraft Wiki; (3) a new agent capability benchmark that encompasses tasks requiring long-term planning, dynamic-immediate planning, and autonomous exploration. The public availability of all datasets, model weights, and code will facilitate future research in the development of autonomous agents. We hope that ODYSSEY will inspire further innovation and progress in the field of autonomous agent development.

**Limitations and Future Works.** The proposed open-world skill library enables the use of open-source LLMs as the foundation for agents to call upon skills, avoiding the high costs associated with previous work using GPT-4 (Wang et al., 2023a; Li et al., 2023; Qin et al., 2023). However, the open-source LLMs are prone to generating hallucinations, leading to a decrease in agent performance. Thus, our future research will focus on employing retrieval-augmented generation to improve LLMs in Minecraft. Additionally, this work focuses on developing and evaluating text-based LLMs in the context of Minecraft, with visual aspects currently out of scope. Looking ahead, we plan to integrate visual understanding into the skill library to enhance the agent capabilities.

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

# Appendix

## Table of Contents

## A    DISCUSSION ON SOCIETAL IMPACTS

When developing autonomous embodied agents within Minecraft, the negative impacts are relatively minimal. Minecraft provides a controlled environment to test these technologies. Concerns include potential over-reliance by players, reducing their exploratory and creative thinking, minor data privacy issues due to the collection of anonymized player data, and possible impacts on game balance, particularly in multiplayer settings. Overall, Minecraft is an ideal experimental platform where these mild negative impacts can be effectively managed.

## B    DISCUSSION ON MIGRATING ODYSSEY TO OTHER DOMAINS

The skill library designed for Minecraft is built with modularity and generalizability in mind, allowing for potential adaptation to other domains such as web navigation (Lai et al., 2024; Lu et al., 2024), robot manipulation (Mosemann & Wahl, 2001; Pedersen et al., 2016; Liang et al., 2023; Singh et al., 2023), robot navigation (Zhou et al., 2024b; Shah et al., 2023), and other game-playing environments (Xu et al., 2024). These skills abstract underlying actions and focus on high-level interactions, allowing them to be adapted to different environments by redefining low-level actions without changing the overall structure of the skill library. Even without direct API access, basic action spaces (e.g., keyboard and mouse operations in games, or movement operations in robotics) can be employed to construct primitive skills. Prior research in robotic manipulation, including CaP (Liang et al., 2023) and ProgPrompt (Singh et al., 2023), demonstrates how primitive skills such as picking and placing objects or opening containers can be built from basic actions. Moreover, we believe that the concept of "skills" should extend beyond code APIs to include knowledge from various sources. For example, handbooks can provide informational segments treated as skills, retrievable by LLMs using techniques like retrieval-augmented generation (Lewis et al., 2020), enhancing decision-making.

To fine-tune the LLaMA-3 model for the Minecraft agent, we crawled the Minecraft Wiki and used a GPT-assisted approach to generate an instruction dataset. Researchers in other domains can replicate this process to create their own instruction datasets. To facilitate this, we have open-sourced our Minecraft Wiki crawler on Github, which can be easily modified to crawl similar Wiki websites for other domains. Additionally, our benchmark tasks evaluate agent performance from three perspectives: long-term planning, dynamic-immediate planning, and autonomous exploration. These dimensions effectively assess the capabilities of open-world autonomous agents. Researchers in other domains can adopt these perspectives to design comprehensive evaluation tasks for their needs.

## C    OPEN-WORLD SKILL-BASED INTERACTIVE AGENT

### C.1    OPEN-WORLD SKILL LIBRARY

#### C.1.1    PRIMITIVE SKILLS

Primitive skills encompass a series of underlying interfaces on top of Mineflayer JavaScript APIs (PrismarineJS, 2023), divided into two main categories: 32 operational skills and 8 spatial skills. In addition to Voyager's 18 operational skills Wang et al. (2023a), 14 operational skills implemented by us are presented as follows:

- `plantSeeds(bot, type)`: Let the agent find the nearest farmland and plant a particular kind of seed.
- `feedAnimals(bot, type, count=1)`: Let the agent find the nearest animals of a particular species and numbers and feed them with the appropriate food.
- `killAnimal(bot, type)`: Let the agent kill a particular kind of animal using the best sword in its inventory.
- `killMonsters(bot, type, count=1)`: Let the agent kill monsters nearby of a particular species and numbers using the best sword in its inventory.
- `cookFood(bot, type, count=1)`: Let the agent cook food of a particular kind and numbers using coal and furnace.

- `eatFood(bot, type)`: Let the agent eat a particular kind of food.
- `equipArmor(bot)`: Let the agent equip the best armor(helmet, chestplate, leggings and boots) in its inventory.
- `equipSword/Pickaxe/Axe/Hoe/Shovel(bot)`: Let the agent equip the best corresponding tool in its inventory.
- `getLogs/PlanksCount(bot)`: Return the number of logs/planks (counted in seven different categories) in the inventory.

Additionally, we pioneer 8 spatial skills that Voyager Wang et al. (2023a) lacks, allowing for environmental interactions based on the agent coordinates. The spatial skills implemented by us are presented as follows:

- `findSuitablePosition(bot)`: Let the agent find the best nearby location for placing devices such as a crafting table or furnace. The block must be `minecraft:air` and at least one adjacent reference block exists.
- `checkAdjacentBlock(bot, types, x, y, z)`: Check blocks adjacent to the block at position (x,y,z). Return true if any of the adjacent blocks match the specified types.
- `checkBlockAbove(bot, type, x, y, z)`: Check block above the block at position (x,y,z). Return true if the above block matches the specified type.
- `checkBlocksAround(bot, type, x, y, z)`: Check blocks around the block at position (x,y,z).Return true if any of the around blocks match the specified type.
- `checkNearbyBlock(bot, types, x, y, z, r)`: Check blocks in a radius around the block at position (x, y, z). Return true if any block within the radius matches the specified types.
- `checkNoAdjacentBlock(bot, types, x, y, z)`: Check adjacent blocks of block at position (x,y,z). Return true if not all adjacent blocks are within the specified types.
- `goto(bot, x, y, z)`: Let the agent go to the corresponding position (x,y,z) until it reaches the destination.
- `getAnimal(bot, type, x, y, z)`: Let the agent attract a particular kind of animal to a particular position (x,y,z) with the appropriate food.

### C.1.2 COMPOSITIONAL SKILLS

All compositional skills are encapsulated by the Mineflayer APIs and the aforementioned primitive skills, while higher-level compositional skills recursively call lower-level ones. Fig. 6 illustrates the nested relationships among the 13 skills required to complete the `mineDiamond` task. We classify all compositional skills into main types as follows:

- `mineX(bot)`: Equip the agent with the appropriate tools and find the nearest specific block to mine it.
- `craftX(bot)`: Let the agent collect the necessary materials and check if the crafting table exists in the inventory (if needed), to craft a specific tool or something.
- `smeltX(bot)`: Let the agent check the furnace and fuel, and smelt the specified materials.
- `collectX(bot)`: Similar to `mineX`, used to collect multiple quantities of a certain item.
- `makeX(bot)`: Similar to `craftX`, used to make food.
- `cookX(bot)`: Similar to `smeltX`, used to cook food.
- `plantX(bot)`: Let the agent check the inventory for seeds, collect them if not present, and plant them in nearby farmland.
- `breedX(bot)`: Let the agent check the inventory for the required corresponding feed, find the nearest two animals, feed them, and facilitate their breeding.
- `killX(bot)`: Let the agent equip the best sword in the inventory, find the nearest specific animal or monster, kill it, and collect the dropped items.
- `placeX(bot)`: Let the agent place an item at its current or a nearby suitable location, and if the item is not in inventory, craft it first.

Additionally, there are several other compositional skills aimed at executing specific behaviors, such as `catchFish`, `hoeFarmland`, `shearSheep`, `takeAndMoveMinecart`.

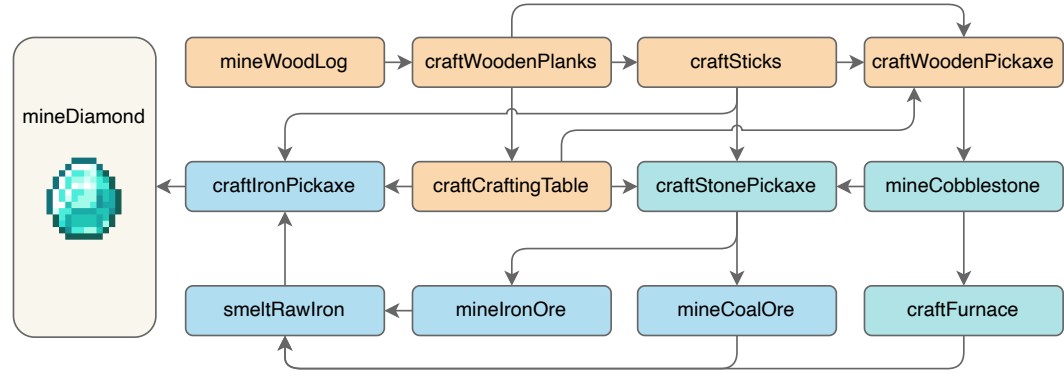

Figure 6: An illustrative diagram of the skill recursive method for the `mineDiamond` task. The four colors depicted represent four different technological levels (wood, stone, iron, and diamond) following the Minecraft tech-tree.

## C.2 LLM PLANNER

ODYSSEY relies on LLMs to generate language-based plans. In our Minecraft experiment, we propose three novel tasks (long-term planning task, dynamic-immediate planning task and autonomous exploration task) for agents to explore. Therefore we designate three types of prompt messages for them respectively, offering LLM Planner the ability to generate different routines on different tasks. The format of the prompt is presented thus:

- "SYSTEM" role: A high-level instruction that gives directions to the model behavior. It sets an overall goal for the interaction and provides external information.
- "USER" role: Detailed information like environment, states and achievements of the agent will be provided to the planner for the next immediate subgoals.
- "ASSISTANT" role: A guideline generated by the planner.

### C.2.1 LONG-TERM PLANNING

We design a suite of combat tasks to assess the long-term planning capabilities of agents, where the LLM Planner should plan to craft appropriate weapons and equipment to defeat monsters.

The input prompt to LLM consists of several components:

- Ultimate goals: The monsters that need to be defeated.
- Directives and behavior constraints that guarantee the proposed task is achievable and verifiable.
- Information of last combat: This ensures that the prompt is exposed to increasing amounts of information over the combat and thus progressively advances towards more efficient plans.

---

**Long-term Planning System Prompt**

*—Overall goals—*

Your goal is to generate the plan that can defeat all monsters while using the shortest time. So, more is not always better when proposing your plan list.

*—External information—*

In Minecraft, combating with monsters requires weapons and armor. The weapon

---

options are limited to "sword", while the armor includes "helmet", "chestplate", "leggings", and "boots". The materials for swords range from low to high level: wooden swords, stone swords, iron swords, and diamond swords; The materials for armor range from low to high level: iron, diamond. The higher the material level, the greater the attack damage of the weapon and the better the protection effect of the armor. However, the higher the material level, the more time it costs to collect.

Tips: Wooden, stone, iron and diamond are the only levels of sword; iron and diamond are the only levels of armors; helmet, chestplate, leggings and boots are the only types of armors; do not generate information that doesn't relate to them.

After each round of combat, I will give you:

**Equipment obtained from last round:** ...
**Health after last combat:** ...
**Critique:** ...
**Monster:** The monsters you need to defeat.

—*Directions*—

The critique (if any) will tell you the subgoal list from the previous round and whether you should trim or add to it. Remember to refer to the critique to adjust your task list. Next, you will start a new combat task, the last round of equipment and health is only for planning reference, not related to the current round. Plan from scratch!

—*Behaviour constraints*—

You must follow the following criteria:
1) Return a Python list of subgoals that can be completed in order to complete the specified task.
2) Each subgoal should only start with "craft"! do not propose any other type of skills!
3) Each subgoal should follow a concise format "craft [material type] [equipment type]".
You should only respond in JSON format as described below:
["subgoal1", "subgoal2", "subgoal3", ...]
Ensure the response can be parsed by Python `json.loads`, e.g.: no trailing commas, no single quotes, etc.

After finish collecting weapons and equipment, we also plan an efficient routine to combat with monsters for higher survival rates. For example, monsters that are more harmful and aggressive should be placed in a higher priority. The full prompt for re-ranking the combat order of monsters is shown below.

**Comabt Order System Prompt**

You are a helpful assistant that generates the order of fighting monsters to defeat all monsters specified by me.
I'll give you a list of monsters, and you need to rearrange the order of monsters according to how hard it is to beat them.
You should give priority to monsters that attack the player and do more damage, while monsters that don't actively attack the player or do less damage should be left behind.
Make sure your list includes all the monsters in your task.
The output format must be exactly same as the input, including the underline.
If your task is to combat a single type of monsters, return a list containing only that monster as well.
You should only respond in JSON format as described below:
["quantity monster1", "quantity monster2", "quantity monster3", ...]
Ensure the response can be parsed by Python `json.loads`, e.g.: no trailing commas, no single quotes, etc.

### C.2.2 DYNAMIC-IMMEDIATE PLANNING

In this kind of task, agents are expected to adapt their plans based on the real-time feedback like nearby resources and animals.

The input prompt to LLM consists of the following components:

- Ultimate goals: A suite of farming tasks, such as planting, harvesting, and animal husbandry.
- The current states of agent: hunger and health values, position and nearby entities, *etc*.
- Achievements of the agent: the current inventory and unlocked equipment, as well as previously successful and failed tasks.

---

**Dynamic-immediate Planning System Prompt**

*—Overall goals—*

You are a helpful assistant that tells me the next immediate task to do in Minecraft. My ultimate goal is to "goals".
Make sure that the proposed task is related to the ultimate goal, and do not propose unrelated tasks!

*—Directions—*

You need to plan step by step towards your ultimate goal, so propose necessary prerequisite tasks first.
For example, "craft hoe" before "hoe farmland", "collect [type] seeds" and "hoe farmland" before "plant seed", "kill [animalType]" before "cook meat", "craft shears" before "shear sheep", "craft bucket" before "collect milk".
Propose the current task only when you ensure that you have all the necessary dependent items in inventory.
Don't ask for repetitive tasks. If you already have an item in your inventory, try not to collect it repeatedly.
For example, when you already have a hoe in your inventory, propose "hoe farmland" instead of "craft hoe" again.

*—External information—*

I will give you the following information:
**Ultimate goal:** ...
**Reference:** ...
**Biome:** ...
**Nearby blocks:** ...
**Other blocks that are recently seen:** ...
**Nearby entities (nearest to farthest):** ...
**Health:** Higher than 15 means I'm healthy.
**Hunger:** Higher than 15 means I'm not hungry.
**Inventory (xx/36):** ...
**Logs:** The execution logs in last task, you can refer to it to propose next task.
**Completed tasks so far:** ...
**Failed tasks that are too hard:** ...

*—Behaviour constraints—*

You must follow the following criteria:
1) Please be very specific about what resources I need to collect, what I need to farm, or what animals I need to breed/kill.
2) The next task should follow a concise format, such as "craft [item]", "breed/kill [animal

---

type]", "cook/eat [food type]", "plant [seed type] seed" or some specific action "shear sheep", "collect milk". Do not propose multiple tasks at the same time. Do not mention anything else.
You should only respond in JSON format as described below:
{
"reasoning": "Based on the information I listed above, do reasoning about what the next task should be",
"task": "The next task"
}
Ensure the response can be parsed by Python `json.loads`, e.g.: no trailing commas, no single quotes, etc.

### C.2.3 AUTONOMOUS EXPLORATION

In this task, the agent is required to explore the Minecraft world freely without any specific goals. This poses a great challenge to the planner for maximal exploration. It should propose suitable tasks based on the current state and environment, e.g., plan to obtain sand or cactus before wood if it finds itself in a desert rather than a forest. The input prompt to LLM consists of several components:

- Guidelines encouraging diverse tasks.
- The current states of agent: hunger and health values, position and nearby entities, *etc*.
- Achievements of the agent: the current inventory and unlocked equipment, as well as previously successful and failed tasks.

---

**Autonomous Exploration System Prompt**

*—Overall goals—*

You are a helpful assistant that tells me the next immediate task to do in Minecraft. My ultimate goal is to discover as many diverse things as possible, accomplish as many diverse tasks as possible and become the best Minecraft player in the world.

*—External information—*

I will give you the following information:
**Biome:** ...
**Time:** ...
**Nearby blocks:** ...
**Other blocks that are recently seen:** ...
**Nearby entities (nearest to farthest):** ...
**Health:** Higher than 15 means I'm healthy.
**Hunger:** Higher than 15 means I'm not hungry.
**Position:** ...
**Equipment:** If I have better armor in my inventory, you should ask me to equip it.
**Inventory (xx/36):** ...
**Chests:** ...
**Completed tasks so far:** ...
**Failed tasks that are too hard:** ...

*—Directions—*

You must follow the following criteria:
1) You should act as a mentor and guide me to the next task based on my current learning progress.
2) Please be very specific about what resources I need to collect, what I need to craft, or what mobs I need to kill.

---

3) The next task should follow a concise format, such as "Mine [block]", "Craft [item]", "Smelt [item]", "Kill [mob]", "Cook [food]", "Equip" etc. It should be a single phrase. Do not propose multiple tasks at the same time. Do not mention anything else.
4) The next task should be novel and interesting. I should look for rare resources, upgrade my equipment and tools using better materials, and discover new things. I should not be doing the same thing over and over again.
5) Don't propose tasks that have already completed once or failed more than three times!
6) Do not ask me to build or dig shelter even if it's at night. I want to explore the world and discover new things. I don't want to stay in one place.
7) Tasks that require information beyond the player's status to verify should be avoided. For instance, "Placing 4 torches" and "Dig a 2x1x2 hole" are not ideal since they require visual confirmation from the screen. All the placing, building and trading tasks should be avoided. Do not propose task starting with these keywords.
8) For wood-related tasks, you don't need to emphasize the type of wood, just propose "mine log" or "craft planks".

—*Behaviour constraints*—

You should only respond in JSON format as described below:
{
"reasoning": "Based on the information I listed above, do reasoning about what the next task should be.",
"task": "The next task."
}
Ensure the response can be parsed by Python `json.loads`, e.g.: no trailing commas, no single quotes, etc.

## C.3 LLM ACTOR

In actor, the mapping from higher language subgoals $S$ to lower executable codes is implemented through query context encoding and similarity retrieval. We employ the following prompt during the generation of query context (Question-Answer pairs).

**Query Context Prompt**

**SYSTEM:**
You are a helpful assistant that answer my question about Minecraft.
I will give you the following information:
Question: ...
You will answer the question based on the context (only if available and helpful) and your own knowledge of Minecraft.
1) Start your answer with "Answer: ".
2) Answer "Answer: Unknown" if you don't know the answer.
**USER:**
How to complete $S$ in Minecraft?

After recalling the top-10 relevant skills with the highest scores, we require LLM to determine the most appropriate code for execution within the environment based on their description. The full prompt of code selection is shown in the following.

**Skill Selection System Prompt**

You are a helpful assistant that decides Mineflayer javascript code to complete any Minecraft task specified by me.
I will give you
**Task:** The task I need to complete in this stage.

**Programs:** The description of relevant programs that may use to complete the task.
**Program used in the last round:** ...
**Critique:** ...

You will choose only one program based on the program description and critique. You should only respond in the format as described below:
{
"program": "your selected program name",
"reason": "Reason you choose the program."
}
Ensure the response can be parsed by Python `json.loads`, e.g.: no trailing commas, no single quotes, etc.
Please ensure that the program name you output should be exactly the same (case-inclusive) as the information provided!

## C.4 LLM CRITIC

The LLM critic should evaluate the success of the executed actions by comparing expected outcomes with actual results, thereby providing valuable critiques for refining strategies in subsequent iterations. We design a chain-of-thought (Wei et al., 2022b) prompting mechanism: We first require LLM to reason about the task's success or failure, then output a boolean variable representing the execution result, and finally provide a critique to the agent if the task fails.

**Critic System Prompt**

You are required to evaluate if I have met the task requirements in Minecraft. Exceeding the task requirements is also considered a success while failing to meet them requires you to provide critique to help me improve.

I will give you the following information:
**Task:** The objective I need to accomplish.
**Nearby blocks:**
**Entities:**
**Equipment:** My tools, weapons and armor could sometimes be here.
**Chests:** If the task requires me to place items in a chest, you can find chest information here.
**Current inventory (xx/36):** My final inventory after carry out the task.
**Last inventory (xx/36):** My inventory before carry out the task.
**Chat log:** The logs during carrying out the task.

**Note** that you only need to check the changes of my inventory to judge whether I meet the task.
For a `craft [item]` task, all you need to do is checking if the item I need to craft is in my current inventory or equipment. If in, you should judge it as a success and vice versa.
For a `mine [item]` task, you only need to check whether the item is in my current inventory or has an increase over the last inventory.
For a `hoe` or `plant` task, you only need to check whether the `farmland` or `seed` is in Nearby Blocks.
Do not judge the success of a `craft` task based on other materials I have!
You can only judge a task failure via chat log, not as a reason to judge a task's success.
You should only respond in JSON format as described below:
{
"reasoning": "reasoning",
"success": boolean,
"critique": "critique",
}

> Ensure the response can be parsed by Python `json.loads`, e.g.: no trailing commas, no single quotes, etc.

The input prompt to LLM consists of the following components:

1. Task proposed by the LLM Planner;

2. Environment feedback: We provide the agent with nearby blocks and entities that are recently seen for high-quality critiques. We also give the log information during the execution stage;

3. Achievements of the agent. We offer achievement of the agent like inventory and equipment to assess the task's completeness.

---

**Critic User Prompt**

Task: Mine 1 wood log
Nearby blocks: birch_leaves, oak_leaves, birch_log, oak_log
Equipment: [None, None, None, None, 'oak_sapling', None]
Chests: None
Current Inventory (2/36): 'oak_sapling': 1, 'oak_log': 1
Last Inventory (0/36):
Chat log: Mined 1 wood log.

---

## D  FINE-TUNE MINECRAFT LLM

For detailed code, datasets, and models used in this section, please visit our code for more information. The overall fine-tuning framework is shown in Fig. 7.

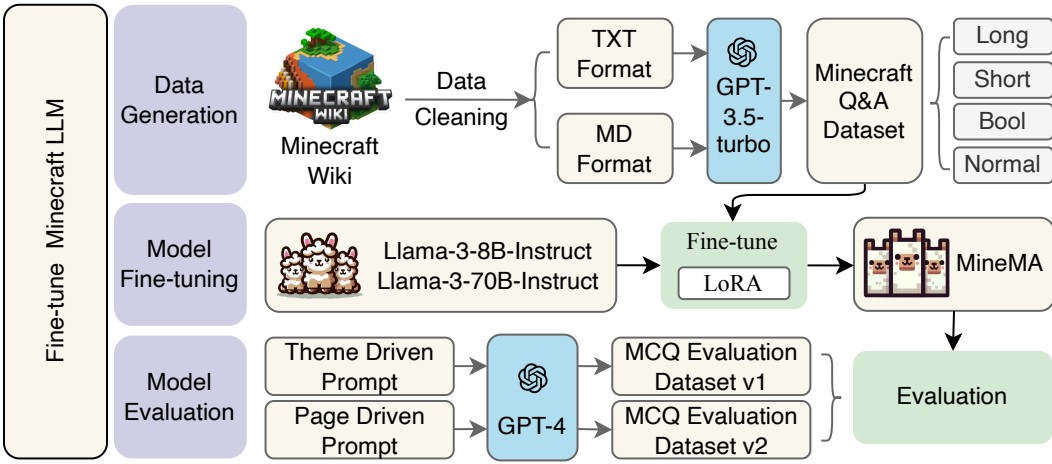

Figure 7: An overview of the fine-tune Minecraft LLM framework.

### D.1  DATASET GENERATION

The code used in this section can be found on the supplementary material. The dataset produced in this section has also been publicly available.

#### D.1.1  DATA CLEANING

For this study, we select two primary sources of information, the Minecraft Fandom Wiki (`https://minecraft.fandom.com/wiki/Minecraft_Wiki`) and the Minecraft Wiki (`https://minecraft.wiki/`).

For the Minecraft Fandom Wiki, we first crawl the content of all its pages and perform a preliminary filtering on the resulting pages, removing pages that were obviously useless for our fine-tuning task, such as game version information, and obtaining a series of JSON files. These files still contain a significant amount of information that we do not need to create the dataset, so we carry out a data cleaning process, extracting the text and table content of the original pages, storing them in a series of TXT files, with each page corresponding to a TXT file. Through the above method, we obtain the cleaned TXT format page information.

For Minecraft Wiki, we exclude a few categories that are useless for our fine-tuning task, such as History, and crawl the content of all other categories' pages. Similar to the process for Minecraft Fandom Wiki, these pages also contain a lot of information that we didn't need. We remove some irrelevant sections such as Achievements, Advancements, and History, and only retained the text and table content of other useful sections. After that, we store the processed data in markdown files and perform segmentation operations based on the number of words in the file. We ensure that the word count of the segmented file did not exceed the set limit without cutting the section block, thus obtaining clean MD format page information.

### D.1.2   Q&A PAIRS

In this study, we design four types of Q&A pairs for our training dataset as follows:

- Short Answer Q&A: Questions that can be directly answered with a word or phrase, such as names, statistics, etc.

- Long Answer Q&A: Questions that require one or more complete sentences as answers. These questions are suitable for explanatory or definitional responses, where a detailed explanation or a clear definition is needed to fully address the question.

- Bool Answer Q&A: Questions that can be answered with a simple True or False.

- Normal Answer Q&A: QA pairs with no specific form restrictions, used to supplement the dataset and ensure its comprehensiveness.

Some Q&A pair examples are shown as follows:

---

**Short Answer Q&A**

Q: What mobs spawn naturally in the desert biome during the day in Minecraft?
A: During the day in Minecraft, zombies in desert biomes are replaced by husks, which inflict the Hunger effect and do not burn in sunlight.

---

**Long Answer Q&A**

Q: How does the use of light sources like torches and lava help in preventing mob spawning in Minecraft?
A: In Minecraft, placing light sources such as torches, lava, glowstone, jack o'lanterns, sea lanterns, and lanterns helps prevent hostile mob spawning. Mobs generally spawn in low light conditions, and these light sources provide enough illumination to raise the light level above the threshold needed for mob spawning. Torches are the most commonly used due to their easy availability and simplicity in creating a well-lit perimeter around an area. Lava is also effective but must be used cautiously as it can ignite nearby flammable materials.

---

**Bool Answer Q&A**

Q: Can skeletons shoot through bamboo?
A: F

---

> **Normal Answer Q&A**
>
> Q: What are piglins in Minecraft?
> A: Piglins are neutral mobs found in the Nether. They become hostile towards players unless the player is wearing at least one piece of golden armor. They can be bartered with using gold ingots for various items, and they wield either a golden sword or a crossbow for melee or ranged attacks, respectively.

### D.1.3 INSTRUCTION DATASETS GENERATION PROMPTS

We design four types of prompts for four types of Q&A pairs to generate instruction datasets by using GPT-3.5-Turbo. Here are our prompts.

**Short Answer Q&A prompt**

> **System Message**
>
> You are a question-and-answer dataset generating expert, you are generating data that will be used to train a large language model designed to address questions posed by users regarding the game Minecraft, and from that, you will generate question-and-answer data samples, each with a prompt/response pair.
> You will do so in this format:
> ```
>
> prompt
> ————
> prompt_goes_here
> ————
> response
> ————
> response_goes_here
> ————
> ```
>
> Your task is to generate at least 30 examples. Make sure your samples are unique and diverse, yet high-quality and complex enough to train a well-performing model.

> **User Message**
>
> Your task is to generate 30 question-and-answer examples, and you should generate questions within the provided user text that can be directly answered with a word or phrase, such as dates, names, statistics, etc. This involves identifying specific, concise information within the text that can be succinctly responded to, ensuring that the answers are clear and directly related to the questions asked. And you will do so in this format:
> ```
>
> prompt
> ————
> prompt_goes_here
> ————
> response
> ————
> response_goes_here
> ————
> ```
>
> Please generate as many question and answer pairs as possible. Here is the user content: {user_content}

**Long Answer Q&A prompt**

**System Message**

You are a question-and-answer dataset generating expert, you are generating data that will be used to train a large language model designed to address questions posed by users regarding the game Minecraft, and from that, you will generate question-and-answer data samples, each with a prompt/response pair.
You will do so in this format:
```

prompt
————

prompt_goes_here
————

response
————

response_goes_here
————
```

Your task is to generate at least 20 examples. Make sure your samples are unique and diverse, yet high-quality and complex enough to train a well-performing model.

**User Message**

Your task is to generate 20 question-and-answer examples. Identify questions within the provided user text that require one or more complete sentences as answers. These questions should be suitable for explanatory or definitional responses, where a detailed explanation or a clear definition is needed to fully address the question. This involves crafting answers that are comprehensive and informative, ensuring they adequately explain or define the subject matter in question. And you will do so in this format:
```

prompt
————

prompt_goes_here
————

response
————

response_goes_here
————
```

Please generate as many question and answer pairs as possible. Here is the user content:
{user_content}

**Bool Answer Q&A prompt**

**System Message**

You are a question-and-answer dataset generating expert, you are generating data that will be used to train a large language model designed to address questions posed by users regarding the game Minecraft, and from that, you will generate question-and-answer data samples, each with a prompt/response pair.
You will do so in this format:
```

prompt
————

prompt_goes_here
————

response
————
```

```
response_goes_here
————————
```
```

Your task is to generate at least 10 examples. Make sure your samples are unique and diverse, yet high-quality and complex enough to train a well-performing model.

---

**User Message**

Your task is to generate 10 question-and-answer examples. Look for questions within the provided user text that can be answered with a simple True or False. This task involves pinpointing statements or queries within the text that lend themselves to binary responses, ensuring that the answers are straightforward and unambiguous, clearly indicating whether the statement is true or false based on the information available. And you will do so in this format:
```

prompt
————————
prompt_goes_here
————————
response
————————
response_goes_here
————————
```
```

Please generate as many question and answer pairs as possible. Here is the user content: {user_content}

**Normal Answer Q&A prompt**

---

**System Message**

You are a question-and-answer dataset generating expert, you are generating data that will be used to train a large language model designed to address questions posed by users regarding the game Minecraft, and from that, you will generate question-and-answer data samples, each with a prompt/response pair.
You will do so in this format:
```

prompt
————————
prompt_goes_here
————————
response
————————
response_goes_here
————————
```
```

Your task is to generate at least 20 examples. Make sure your samples are unique and diverse, yet high-quality and complex enough to train a well-performing model.

---

**User Message**

Your task is to generate 20 question-and-answer examples. And you will do so in this format:
```

prompt
————————
prompt_goes_here
```

---
response
---
response_goes_here
---
```

Please generate as many question and answer pairs as possible. Here is the user content:
{user_content}

## D.2 MODEL FINE-TUNING

The code used in this section can be found on the supplementary material. MineMA-8B and MineMA-70B series of models have also been publicly available .

In this study, we use the instruction dataset with 390,317 instruction entries mentioned above to fine-tune the Minecraft Q&A expert models, using the LoRA fine-tuning method. We name the series of fine-tuned models MineMA. The resulting models include MineMA-8B-v1, MineMA-8B-v2, MineMA-8B-v3, derived from the base model LLama-3-8B-Instrument, and MineMA-70B-v1, MineMA-70B-v2, derived from the base model LLama-3-70B-Instrument. MineMA-70B series of models are fine-tuned on four A6000 GPUs, while the remaining models are fine-tuned on a single A6000 GPU each. Among the models, MineMA-8B-v1 and MineMA-70B-v1 only undergo one round of training without an evaluation process, while the other models are trained with multiple rounds that incorporate an evaluation procedure. We use the EarlyStopping method to halt the training process when there is no reduction in the evaluation loss over a series of evaluations, and finally save the model which has the best performance. Some training parameters are shown in Tab. 5.

Table 5: Training parameters for different MineMA models.

| Model | LoRA r | LoRA alpha | LoRA dropout | Learning Rate | Weight Decay | Single Round? |
|---|---|---|---|---|---|---|
| MineMA-8B-v1 | 64 | 128 | 0.1 | 1E-04 | 1E-04 | T |
| MineMA-8B-v2 | 32 | 64 | 0.1 | 1E-04 | 1E-04 | F |
| MineMA-8B-v3 | 64 | 128 | 0.1 | 1E-04 | 1E-04 | F |
| MineMA-70B-v1 | 16 | 32 | 0.1 | 1E-04 | 1E-04 | T |
| MineMA-70B-v2 | 64 | 128 | 0.1 | 1E-04 | 1E-04 | F |

## D.3 MODEL EVALUATION

The code used in this section can be found on the supplementary material. The datasets used in this section have also been publicly available.

### D.3.1 EVALUATION DATASETS CREATING PROCESS

In this study, we utilize GPT-4 to create two evaluation MCQ datasets: a multi-theme MCQ dataset and a Wiki-based MCQ dataset. For the multi-theme MCQ dataset, we first summarize the following Minecraft content themes:

**Game Basics**

Blocks and Items: Basic blocks, special blocks, tools, weapons, armor, etc.
Survival Mechanics: Health, hunger, experience levels, death and respawn, etc.

**World Exploration**

Biomes: Characteristics of different biomes, generated structures, unique resources, etc.
Terrain and Landforms: Features and resource distribution of different terrains.

**Mobs and Interactions**

Mobs: Characteristics and behaviors of passive, neutral, and hostile mobs.
Combat System: Monster types, combat tactics, weapons and equipment, enchantments, potions, etc.
Trading and Villagers: Villager professions, trading mechanics, village structures, etc.

**Survival Skills**

Resource Gathering: Methods of obtaining various resources and their uses.
Crafting and Production: Usage of crafting tables, furnaces, etc., equipment crafting and upgrading.
Farming and Animal Husbandry: Crop planting, animal breeding, automated farms, etc.

**Building and Creativity**

Building Styles: Various building styles and key points.
Building Techniques: Symmetry, proportion, detail handling in construction, etc.
Interior Decoration: Interior design, lighting, item placement, etc.
Redstone Mechanics: Redstone components, circuit design, automated devices, etc.

**Special Dimensions**

The Nether: Peculiarities of the Nether, unique blocks and mobs, special structures, etc.
The End: Characteristics of the End, Ender Dragon, cities, ships, etc.
Adventure and Exploration: Special generated structures like ocean monuments, woodland mansions, ruins, fortresses, etc.

Then, we list different numbers of keywords for each theme based on the amount of relevant knowledge content. According to the amount of information related to each keyword, we match a number for each keyword, representing the number of multiple-choice questions to be generated based on that keyword. After preparing the groundwork, we use GPT-4 to generate the multi-theme MCQ dataset, totaling 1,050 multiple-choice questions. The relevant prompts are shown below, taking the generation of multiple-choice questions in the Special Dimensions theme as an example:

**System Message**

You are an expert in generating Minecraft quiz questions. Your task is to create multiple-choice questions about the game Minecraft based on the theme of "Special Dimensions" and the provided keywords. The introduction of the theme of "Special Dimensions" is as follows:
Special Dimensions:
The Nether: Peculiarities of the Nether, unique blocks and mobs, special structures, etc.
The End: Characteristics of the End, Ender Dragon, cities, ships, etc.
Adventure and Exploration: Special generated structures like ocean monuments, woodland mansions, ruins, fortresses, etc.
Provide four answer options labeled A, B, C, and D. Only one option should be correct.
After the question and options, state the correct answer. Please format the output as follows:
Difficulty: Easy/Medium/Hard
Topic: Special Dimensions
Key Word: text
Question: Question text
Options: A.text B.text C.text D.text
Correct Answer: A/B/C/D
Ensure that the difficulty distribution of the questions and options is reasonable, and the answers should be detailed and informative.

**User Message**

Please generate some Minecraft multiple-choice questions based on the following 5 keywords, covering three difficulty levels: simple, moderate, and difficult. The number after the keyword represents how many multiple-choice questions to generate based on this keyword.
Keywords:
{keywords_go_here}
Ensure that the Q&A content is rich and accurate, and test the player's understanding of the game. Provide a balanced combination of simple, medium, and difficult questions. Generate each question and answer in the given format. Here is an example:
Example:
Difficulty: Hard
Topic: Special Dimensions
Key Word: End Ship
Question: What exclusive item can be found in the End Ship in Minecraft?
Options: A. Netherite B. Dragon Egg C. Elytra D. Beacon
Correct Answer: C

For the Wiki-based MCQ dataset, we utilize GPT-4's knowledge of Minecraft-related Wiki content to create a set of multiple-choice questions that closely align with the information on the Wiki pages. Firstly, we list 615 Minecraft-related keywords based on the importance of the relevant knowledge. Afterwards, we generate a Wiki-based MCQ dataset using GPT-4 with designed prompts based on these keywords, totaling 2,083 pieces of data. The prompts we used are as follows:

**System Message**

You are an expert in generating Minecraft multiple-choice questions. Your task is to create multiple choice questions about the game Minecraft based on the provided keywords and the information on the corresponding page on the Minecraft Wiki. Ensure that the source of information for the multiple-choice questions you generate is the Minecraft Wiki, while ensuring the objectivity and accuracy of the multiple-choice questions and ensuring good quality.
Provide four answer options labeled A, B, C, and D. Only one option should be correct. After the question and options, state the correct answer. Please format the output as follows:
Difficulty: Easy/Medium/Hard
Key Word: text
Question: Question text
Options: A.text B.text C.text D.text
Correct Answer: A/B/C/D
Ensure that the difficulty distribution of the questions and options is reasonable, and the answers should be detailed and informative.

**User Message**

Please generate some Minecraft multiple-choice questions based on the following 5 keywords, covering three difficulty levels: simple, moderate, and difficult. The number after the keyword represents the minimum number of multiple-choice questions generated based on the keyword. For important keyword, you should generate more questions.
Keywords:
{keywords_go_here}
Ensure the source of information for the multiple-choice questions you generate is the Minecraft Wiki, while ensuring the objectivity and accuracy of the multiple-choice questions and ensuring good quality. Provide a balanced combination of simple, medium, and difficult questions. Generate each question and answer in the given format, do not use '#'or ".. Here is an example:
Example:
Difficulty: Medium

> Key Word: Dirt
> Question: What happens when you right-click on a dirt block with a hoe?
> Options: A. It turns into farmland B. It turns into grass C. It turns into a path block D. Nothing happens
> Correct Answer: A

### D.3.2 EVALUATION RESULTS

We use the above two MCQ datasets to evaluate the MineMA series models and the corresponding base models. Each model is tested 5 times with the two datasets. The results are shown in Tab. 6.

Table 6: The evaluation results based on the MCQ datasets.

| Model | Average Accuracy (Multi-theme) | Average Accuracy (Wiki-based) |
|---|---|---|
| Llama-3-8b-Instruct | 61.09% | 54.38% |
| MineMA-8B-v1 | 62.69% | 61.97% |
| MineMA-8B-v2 | 62.23% | 62.09% |
| MineMA-8B-v3 | 62.99% | 62.42% |
| Llama-3-70b-Instruct | 77.41% | 72.52% |
| MineMA-70B-v1 | 78.11% | 73.03% |
| MineMA-70B-v2 | 75.68% | 72.88% |

## E  AGENT CAPABILITY BENCHMARK

### E.1  LONG-TERM PLANNING TASK

In Minecraft, there are a total of 35 hostile creatures. We conducted experiments on both single-monster combat tasks and combined combat tasks (up to three types of monsters), resulting in thousands of different tasks that can all be implemented through the interfaces we provided.

- `combatEnv(bot, h, r, y)`: Generates a hollow rectangular arena with a height of $h$ and a square base with side length $2r$ at altitude $y$, positioning the agent at the exact center of this enclosed space. This configuration ensures controlled conditions for evaluating combat scenarios, especially considering not being influenced by naturally spawning monsters.
- `summonMob(bot, n = 1, r, type)`: Facilitates the spawning of hostile creatures around the bot. It randomly positions $n$ monsters within a designated range ($r$ to $2r$ along the $x$ and $z$ axes) from the bot, allowing for the creation of varied combat tasks and enabling comprehensive testing of bot performance under different tactical challenges.

### E.2  DYNAMIC-IMMEDIATE PLANNING TASK

In Minecraft, many farming tasks require interaction with the environment and dynamic planning. We propose a series of tasks that can be accomplished through our skill library, including hoeing farmland, planting seeds, harvesting crops, making food, slaughtering animals, cooking meat, feeding and breeding animals, among others. For example, in the task `cook meat`, if the agent is informed that there is a chicken nearby, it should plan to "kill one chicken" rather than anything else. Additionally, in the task `milk cow`, the agent must simultaneously monitor the appearance of cows in the vicinity and gather materials to craft a bucket to collect the milk.

### E.3  AUTONOMOUS EXPLORATION TASK

In Minecraft, autonomous exploration is the gameplay approach that most closely mimics how human players engage with the game. To evaluate the diversity of discoveries made by the agent during autonomous exploration, we used "Distinct Items Obtained" as the primary evaluation metric. The acquisition of a greater variety of items demonstrates more diverse exploratory behavior

by the agent. Additionally, based on statistical information and progress in-game achievements, we calculated supplementary evaluation metrics including the "Distance Traveled" by the agent (summing walking, sprinting, climbing, swimming, and other forms of movement), the total number of "Items Crafted" (the sum of all types of items obtained by crafting), and "Recipes and Achievements Unlocked" (the sum of crafting recipes and game achievements unlocked).

### E.4 SPECIFIC AGENT CAPABILITY REQUIREMENTS FOR DIFFERENT TASKS

This section provides an overview of the specific agent capabilities required for each task, laying the foundation for a deeper understanding of how our benchmark evaluates different aspects of agent performance. Different agent capabilities are detailed as follows:

- **Goal-based Planning:** This capability refers to the agent's ability to formulate and execute comprehensive plans based on predefined goals. It involves understanding the given goals and devising a step-by-step plan to achieve them over extended periods. This is critical for tasks such as the long-term planning task, where agents need to craft weapons and equipment to defeat specific monsters.

- **Feedback-based Planning:** This capability involves the agent's ability to adapt its plans dynamically based on environmental feedback. It is essential for tasks where environmental feedback is crucial, such as in the dynamic-immediate planning task and the multi-round long-term planning task, where agents must adjust their strategies in response to the outcomes of previous actions or environmental changes.

- **Exploratory Planning:** This capability evaluates the agent's ability to set its own goals and make decisions independently in a complex environment. Agents must navigate, gather information, and decide on objectives without predefined goals. This is central to the autonomous exploration task, where agents explore the Minecraft world, discover resources, and adapt to unforeseen events.

- **Task Decomposition:** This capability refers to the agent's ability to break down complex tasks into specific, manageable sub-tasks. It is vital for the long-term planning task where agents need to craft a sequence of items, requiring the breakdown of the end goal into a series of intermediate steps.

- **Resource Management:** This capability involves the efficient allocation and utilization of available resources. Agents must maintain awareness of their inventory, manage assets effectively, and identify which resources need to be gathered. This is particularly important in farming tasks and autonomous exploration, where resource availability and management are crucial for subsequent behavior.

- **Skill Retrieval:** This capability pertains to the agent's ability to identify and choose the most suitable skill from a set of options. Agents evaluate a list of relevant skills and select the one that best fits the current environmental context and task requirements. All tasks require agents to retrieve and apply relevant skills based on situational demands.

- **Self-Reflection:** This capability involves the agent's ability to analyze and learn from the outcomes of its actions. Simply confirming the completion of a subgoal is often inadequate for correcting planning errors. The agent evaluates its performance, deduces the cause of task failures, and suggests more efficient strategies for future tasks. This is particularly important in multi-round tasks.

- **Self-Validation:** This capability enables the agent to autonomously confirm the success of its actions against intended outcomes. By assessing inventory changes after actions, the agent ensures that each step contributes towards the overarching objectives without external verification. This capability is crucial for all tasks, as agents need to continuously ensure their actions align with the objectives.

## F EXPERIMENTS

### F.1 EXPERIMENTAL DETAILS

We select the 1.19.4 version of Minecraft as the experimental environment. Within this virtual game world, spatial measurements are determined by blocks, while temporal measurements are dictated

by ticks, each lasting 0.05 seconds. A single day-night cycle in the game is 24,000 ticks, equivalent to 20 minutes in the real world, with 10 minutes of daytime, 7 minutes of nighttime, and a 3-minute dawn/dusk transition (when both the sun and moon are visible in the sky). Additionally, the game's weather system randomly transitions between clear, rainy, thunderstorm, and snowy conditions, adding dynamic changes to the environment. Players are born into a randomly generated massive world, covering an area of 30,000,000 blocks × 30,000,000 blocks, which can be approximately considered an infinite world without boundaries. Players start with no resources and must gather everything from scratch that is beneficial for survival and completing the ultimate goal. When a player character dies, it will respawn randomly within a 32-block radius of the death location on the ground, and any collected items will not be dropped. Agents can connect to the game through local networks or multiplayer servers. We have tested on Ubuntu 20.04, Windows 10, and macOS. In all experiments of the agent capability benchmark, the "MineMA-8B" refers to "MineMA-8B-v3", and the "MineMA-70B" refers to "MineMA-70B-v1".

We use the following Minecraft mods in our experiment. It is important to note that the version of mods must be consistent with the game version, specifically 1.19.4.

- **Fabric API:** Basic Fabric APIs.
- **Mod Menu:** Used to manage all the mods that you download.
- **Complete Config:** Dependency of server pause.
- **Multi Server Pause:** Used to pause the server when waiting for LLM to reply.
- **Better Respawn:** Used for random respawning of player characters.

Considering the randomness of resource distribution in the Minecraft world, we ensure that the agent starts from different locations in the game world before each round of experiments. We implemented the `respawnAndClear` interface to perform some initialization settings.

- `respawnAndClear(bot)`: Transport the agent to a new location and clear its inventory, ensuring that the game mode is switched to survival and the game difficulty is switched to peaceful.

### F.2  AGENT CAPABILITY BENCHMARK

In our multi-round **Long-term Planning Task**, the agent is required to iteratively improve planning based on combat outcomes, aiming for victory with the highest efficiency, take as little time as possible. Specifically, if the agent wins in the previous round, it should streamline its planning in the next round, gathering materials and crafting equipment in less time to enhance time efficiency (reflected in the experimental results as a decrease in time and LLM iterations); conversely, if it loses, it must refine its planning to upgrade the quality of weapons and equipment in the planning list to ensure ultimate success (reflected in the experimental results as an increase in health, or go from losing to winning). Additionally, when calculating experimental results, we compute the average and standard deviation for time, LLM iters (LLM iterations) and the health metric only for victorious outcomes, since a defeat, indicated by health being zero, is not meaningful.

---

**Example of multi-round combat task**

Combat Task: 1 Skeleton

Plan list of 1st round:[craft iron sword, craft iron helmet, craft iron chestplate, craft iron leggings, craft iron boots]
Equipment obtained of 1st round: [iron_helmet, iron_chestplate, iron_leggings, iron_boots, crafting_table, None]
Time spent on crafting equipment: 15,953 ticks; 8 LLM iters
Remaining Health after the combat: 14.0 / 20 (victory)

—*streamlining*—

Plan list of 2nd round:[craft iron sword]
Equipment obtained of 2nd round:[None, None, None, None, iron_sword, None]
Time spent on crafting equipment: 3,614 ticks; 4 LLM iters

---

> Remaining Health after the combat: 9.2 / 20 (victory and more efficiently)
>
> —*streamlining*—
>
> Plan list of 3rd round:[craft wooden sword]
> Equipment obtained of 3rd round:[None, None, None, None, wooden_sword, None]
> Time spent on crafting equipment: 416 ticks; 1 LLM iter
> Remaining Health after the combat: 9.0 / 20 (victory and even more efficiently)

In our **Dynamic-immediate Planning Task**, the agent is asked to plan step by step based on environmental information. We calculate the success rate across various tasks, the average execution time and LLM iters as well as their standard deviation (only if successful) as evaluation metrics. It is important to note that skills used in these tasks do not utilize the recursive decomposition mechanism we propose but require the agent to plan the necessary preparatory steps by itself. The following outlines the specific skill execution pathways for the five tasks in our experiments:

> **Skill execution path of the Dynamic-immediate Planning Task**
>
> **Collect Seeds**: Collect Wheat Seeds / Collect Melon Seeds / Collect Pumpkin Seeds
> **Hoe Farmland**: Craft Hoe → Hoe Farmland
> **Shear Sheep**: Craft Shears→Shear Sheep Using Shears
> **Milk Cow**: Craft Bucket→Milk Cow Using Bucket
> **Cook Meat**: Kill Pig→Cook Porkchop / Kill Chicken→Cook Chicken / Kill Sheep→Cook Mutton / Kill Cow→Cook Beef

In our **Autonomous Exploration Task**, the agent also needs to plan step by step without a given goal. Every time a new plan is proposed, the agent retrieves the ten most semantically similar skills from our skill library and selects one to execute. We tally the number of distinct item types obtained by the agent in each round, as well as the cumulative number of item types. Here are the distinct items obtained by the agent from one round of the experiment:

> **Distinct items obtained within 80 LLM iters**
>
> ['oak_log', 'stick', 'wooden_sword', 'crafting_table', 'wooden_pickaxe', 'stone_pickaxe', 'oak_planks', 'wheat_seeds', 'dirt', 'cobblestone', 'raw_iron', 'granite', 'andesite', 'cobbled_deepslate', 'diorite', 'diamond', 'iron_pickaxe', 'furnace', 'cobblestone_wall', 'coal', 'iron_ingot', 'iron_trapdoor', 'dandelion', 'azure_bluet', 'poppy', 'oxeye_daisy', 'chest', 'cobblestone_stairs', 'raw_copper', 'copper_ingot', 'calcite', 'copper_block', 'birch_planks', 'jungle_log', 'arrow', 'bone', 'rotten_flesh'], Num: 37

This result is comparable to the Voyager Wang et al. (2023a) framework that employs GPT-4 for skill code generation and significantly outperforms Voyager using GPT-3.5.

F.3 ABLATION STUDY

We conduct ablation studies on two core components of the ODYSSEY agent, including the LLM planner and the open-world skill library.

For the LLM planner ablation, we remove the current environmental information in the planner system prompt as follows. Moreover, in each task proposed during each round, if the retrieved skills were not relevant to the current task (i.e., if the semantic retrieval score was below a certain threshold), the execution of those skills was not carried out.

> **Planner System Prompt in Ablation**
>
> You are a helpful assistant that tells me the next immediate task to do in Minecraft. My ultimate goal is to discover as many diverse things as possible, accomplish as many diverse

tasks as possible and become the best Minecraft player in the world. You can propose next suitable tasks for me, such as "Mine [block]", "Craft [item]", "Smelt [item]", "Kill [mob]", "Cook [food]", "Equip" etc. It's better to be a single phrase.

You should only respond in JSON format as described below:
{
"reasoning": "Do reasoning about what the next task should be.",
"task": "The next task."
}

Ensure the response can be parsed by Python `json.loads`, e.g.: no trailing commas, no single quotes, etc.

For the open-world skill library ablation, we removed the entire skill library and provided the LLM only with the necessary interfaces required for composing new skills. Each round's skill retrieval and execution were changed to code writing and execution, similar to the approach used in Voyager Wang et al. (2023a). The actor system prompt is shown as follows:

---

**Actor System Prompt in Ablation**

You are a helpful assistant that writes Mineflayer javascript code to complete any Minecraft task specified by me.

—*External information*—

At each round of conversation, I will give you
**Code from the last round:** ...
**Execution error:** ...
**Chat log:** ...
**Biome:** ...
**Nearby blocks:** ...
**Nearby entities (nearest to farthest):**
**Health:** ...
**Hunger:** ...
**Position:** ...
**Equipment:** ...
**Inventory (xx/36):** ...
**Chests:** ...
**Task:** ...
**Context:** ...
**Critique:** ...

—*Directions*—

You should then respond to me with
**Explain** (if applicable): Are there any steps missing in your plan? Why does the code not complete the task? What does the chat log and execution error imply?
**Plan**: How to complete the task step by step. You should pay attention to Inventory since it tells what you have. The task completeness check is also based on your final inventory.
**Code**:
1) Write an async function taking the bot as the only argument.
2) Reuse the above useful programs as much as possible.
3) ...

—*Behaviour constraints*—

You should only respond in the format as described below:

---

```
Explain: ...
Plan:
1) ...
2) ...
3) ...
...
Code:
```javascript
// helper functions (only if needed, try to avoid them)
...
// main function after the helper functions
async function yourMainFunctionName(bot) {
// ...
}
```
```

## F.4 RESULTS

We compare smaller LLMs (LLaMA-3-1B and LLaMA-3-3B) on the single-round long-term planning tasks. The results in Table 7 show that larger model parameters lead to better agent performance. Specifically, LLaMA-3-3B generally performs better than LLaMA-3-1B, achieving higher success rates across the tasks. Moreover, we additionally provide Figure 8 and Figure 9 displaying the results of the single-round long-term planning task and the dynamic-immediate planning task for easier visual inspection.

Table 7: Performance comparison of smaller LLMs on the single-round long-term planning task. All evaluation metrics are calculated only for successful tasks. $\pm$ corresponds to one standard deviation of the average evaluation over successful tasks.

| Task | Model | Success Rate | Health | Time (min) | # LLM Iters |
|---|---|---|---|---|---|
| 1 zombie | MineMA-8B | 8 / 8 | $19.4 \pm 2.3$ | $8.8 \pm 5.4$ | $10.0 \pm 5.8$ |
| | LLaMA-3-8B | 4 / 8 | $20.0 \pm 0.0$ | $8.3 \pm 4.2$ | $6.1 \pm 4.1$ |
| | LLaMA-3-3B | 4 / 8 | $20.0 \pm 0.0$ | $19.4 \pm 5.3$ | $12.5 \pm 5.1$ |
| | LLaMA-3-1B | 2 / 8 | $20.0 \pm 0.0$ | $9.4 \pm 2.7$ | $9.5 \pm 4.9$ |
| 1 spider | MineMA-8B | 8 / 8 | $19.3 \pm 1.6$ | $8.3 \pm 6.7$ | $15.2 \pm 6.0$ |
| | LLaMA-3-8B | 4 / 8 | $19.4 \pm 1.0$ | $12.1 \pm 3.8$ | $8.4 \pm 3.5$ |
| | LLaMA-3-3B | 3 / 8 | $18.1 \pm 3.3$ | $9.1 \pm 2.8$ | $9.7 \pm 5.7$ |
| | LLaMA-3-1B | 2 / 8 | $19.5 \pm 0.7$ | $9.8 \pm 1.3$ | $8.5 \pm 6.4$ |
| 1 skeleton | MineMA-8B | 8 / 8 | $13.6 \pm 5.9$ | $8.6 \pm 7.3$ | $12.1 \pm 7.0$ |
| | LLaMA-3-8B | 4 / 8 | $17.6 \pm 2.7$ | $8.1 \pm 3.5$ | $8.9 \pm 3.7$ |
| | LLaMA-3-3B | 4 / 8 | $18.5 \pm 1.5$ | $11.5 \pm 7.2$ | $12.5 \pm 6.4$ |
| | LLaMA-3-1B | 3 / 8 | $19.6 \pm 0.5$ | $14.0 \pm 8.6$ | $10.3 \pm 9.2$ |

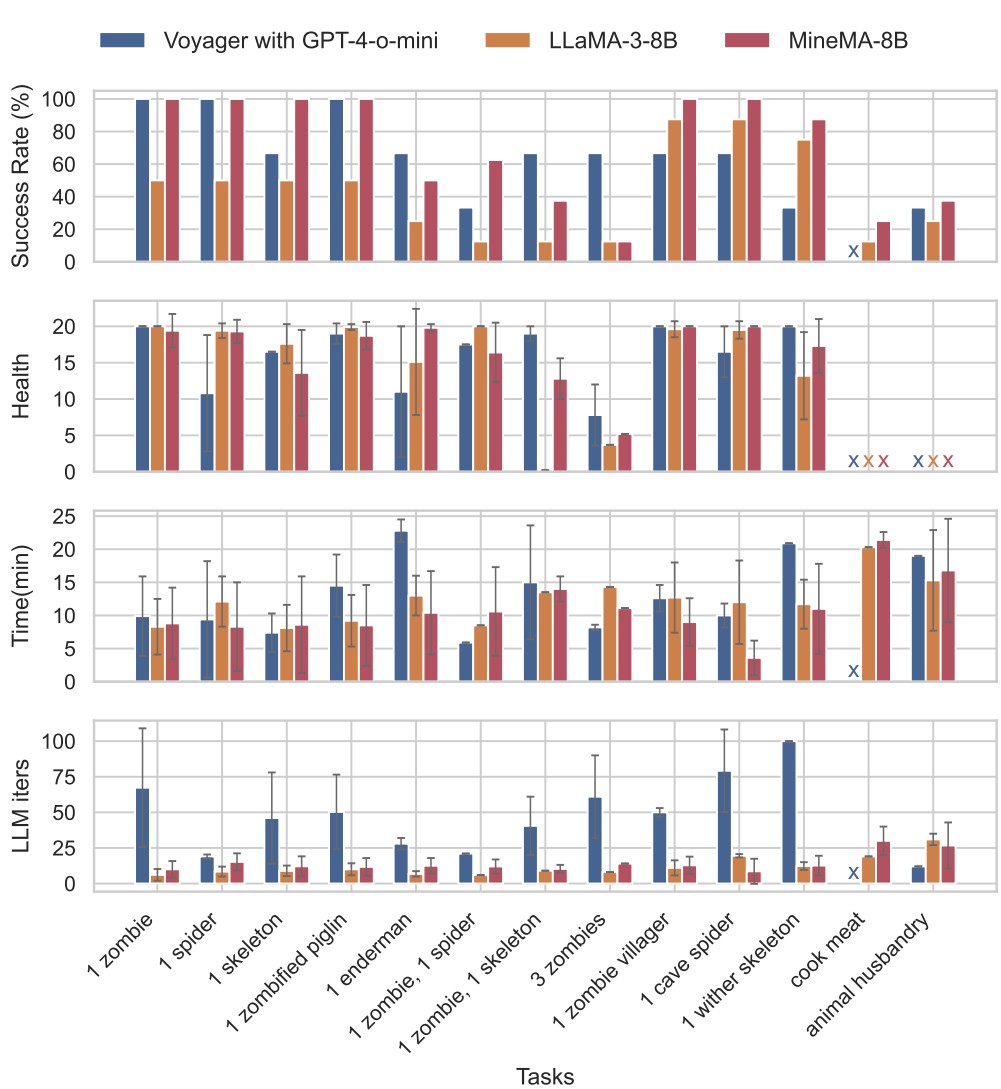

Figure 8: Performance comparison of different models on the single-round long-term planning task. "Health" refers to the remaining health points. "# LLM iters" is the number of LLM iterations (calling LLM) required to complete the task. "Time (min)" refers to the minutes spent in both gathering materials and crafting equipment to defeat different monsters. All evaluation metrics are calculated only for successful tasks. $\pm$ corresponds to one standard deviation of the average evaluation over successful tasks. **Bold** and *italics* mean the best and the second-best results. "x" indicates that health is not a relevant metric in the *cook meat* and *animal husbandry* scenarios, or all tasks fail.

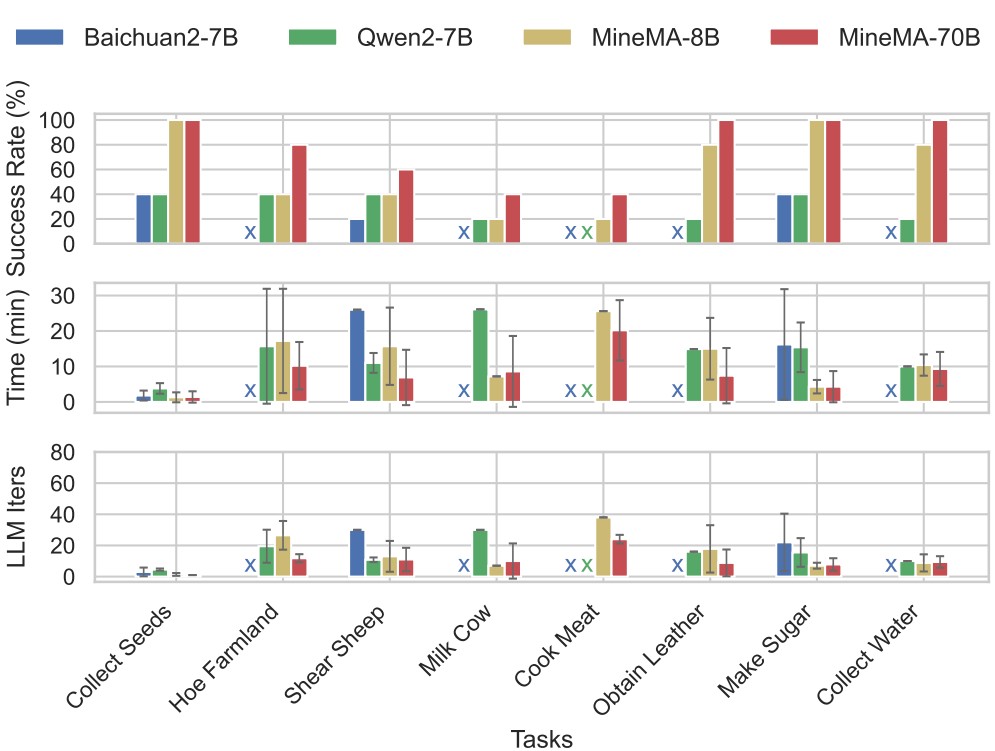

Figure 9: Performance comparison of different models on the dynamic-immediate planning task. All evaluation metrics are calculated only for successful tasks. "x" indicates that all tasks fail.

