# OpenReview forum: "Odyssey: Empowering Minecraft Agents with Open-World Skills"
_ICLR.cc/2025/Conference — ICLR 2025 Conference Withdrawn Submission_

### Official Review · Reviewer_bSai · 2024-10-26

**Soundness:** 3
**Presentation:** 3
**Contribution:** 3
**Rating:** 6
**Confidence:** 5

**Summary:**

There is a growing interest in using LLMs as generalist agents for open-world decision-making settings like the video game Minecraft. The authors demonstrate by example that even moderately sized LLMs (~8B parameters) are capable of performing well in this video game when (1) fine-tuned on a large question-answering dataset specific to the domain and (2) interfaced with a rich, hand-engineered skill library. Applying these ingredients to the Llama 3 8B parameter LLM, the authors show that it is possible to achieve performance that is on par with a Voyager GPT-4o Minecraft agent. The authors open source their datasets, model weights, and code.

**Strengths:**

- The paper is polished and well-written.
- Experiments and analyses of results are thorough. Models that are trained and evaluated using the proposed framework are compared against relevant baselines.
- The code released by the authors is clean and easy to use.
- The performance of LMs under agentic frameworks like Voyager, which prompt models to generate skill libraries as code from scratch, depends strongly on the ability of the base model to generate quality code. In contrast, the Odyssey framework enables future work studying "tool use" in Minecraft *across* LM parameter scales by decoupling the evaluation of LMs as "high-level" vs "low-level" agentic controllers. This is a valuable contribution to the community.

**Weaknesses:**

- The proposed framework has limited novelty. Decomposing complex decision-making tasks with hand-engineered skill libraries has a very long history in robotics [1,2].
- The Odyssey framework is designed specifically for Minecraft. Agentic performance is significantly boosted through the careful design of useful, hand-engineered low-level skills. As a result, it is unclear to what extent good LM performance on Minecraft with Odyssey would transfer to other, more practical open-world environments like Web navigation.



[1] Mosemann, Heiko, and Friedrich M. Wahl. "Automatic decomposition of planned assembly sequences into skill primitives." IEEE transactions on Robotics and Automation 17.5 (2001): 709-718.
[2] Pedersen, Mikkel Rath, et al. "Robot skills for manufacturing: From concept to industrial deployment." Robotics and Computer-Integrated Manufacturing 37 (2016): 282-291.

**Questions:**

It would be interesting how well even smaller LMs than Llama 3 8B would perform on Minecraft under the Odyssey framework. Have any experiments of this sort been conducted?

---

> ### Author Response · Authors · 2024-11-23
> **Response (1/2)**
>
> We sincerely appreciate your positive feedback on Odyssey, particularly highlighting its valuable contribution to the community. We have carefully revised the manuscript according to your valuable suggestions. Below we address the main points raised in the review.
>
> ---
>
> **W1. The proposed framework has limited novelty. Decomposing complex decision-making tasks with hand-engineered skill libraries has a very long history in robotics [1,2].**
>
> Thanks for your valuable comments. (1) We would like to emphasize that our contribution does not lie in technical novelty. This is why we select "datasets and benchmarks" as the primary area in the OpenReview submission, which is more suited for contributions like ours that offer new datasets, benchmarks, and tools to the research community. We believe that one of our significant contributions is indeed the engineering domain-specific skill library, which represents a meaningful addition to the Minecraft agent community. Developing this library requires significant time and effort in design and debugging, making it a valuable resource for researchers. By leveraging our skill library, even open-source LLMs can enable open-world exploration within the Minecraft environment.
>
> > LLM-based Voyager advances exploration using skill generation but relies on GPT-4, incurring substantial costs (reaching thousands of dollars per experiment). This expense can be prohibitive for many researchers. On the contrary, our agent  can call upon the proposed skill library, achieving performance comparable to Voyager with GPT-4, but using only 8B LLMs.
>
> (2) The skill library is not the only contribution of our work. The proposed Odyssey consists of three key contributions: (a) a fine-tuned LLaMA-3 model trained on a large-scale question-answering dataset; (b) an interactive agent equipped with an open-world skill library; (c) a new agent capability benchmark encompassing a variety of tasks. We have open-sourced all parts of ODYSSEY (see supplementary material) and will continuously update the repository. We hope this will enable other researchers to build upon our work, fostering further innovation and progress in the development of autonomous agents.
>
>
> **W2. The Odyssey framework is designed specifically for Minecraft. Agentic performance is significantly boosted through the careful design of useful, hand-engineered low-level skills. As a result, it is unclear to what extent good LM performance on Minecraft with Odyssey would transfer to other, more practical open-world environments like Web navigation.**
>
>
> Thanks for your insightful comments. We have provided a discussion on migrating Odyssey to other domains in the revision (**Page 16, Appendix B**), as follows:
>
>
> > Discussion on Migrating Odyssey to Other Domains
>
> > The skill library designed for Minecraft is built with modularity and generalizability in mind, allowing for potential adaptation to other domains such as web navigation [3, 4], robot manipulation [1, 2, 5, 6], robot navigation [7, 8], and other game-playing environments [9]. These skills abstract underlying actions and focus on high-level interactions, allowing them to be adapted to different environments by redefining low-level actions without changing the overall structure of the skill library. Even without direct API access, basic action spaces (e.g., keyboard and mouse operations in games, or movement operations in robotics) can be employed to construct primitive skills. Prior research in robotic manipulation, including CaP [5] and ProgPrompt [6], demonstrates how primitive skills such as picking and placing objects or opening containers can be built from basic actions. Moreover, we believe that the concept of "skills" should extend beyond code APIs to include knowledge from various sources. For example, handbooks can provide informational segments treated as skills, retrievable by LLMs using techniques like retrieval-augmented generation [10], enhancing decision-making.
>
> > To fine-tune the LLaMA-3 model for the Minecraft agent, we crawled the Minecraft Wiki and used a GPT-assisted approach to generate an instruction dataset. Researchers in other domains can replicate this process to create their own instruction datasets. To facilitate this, we have open-sourced our Minecraft Wiki crawler on Github, which can be easily modified to crawl similar Wiki websites for other domains. Additionally, our benchmark tasks evaluate agent performance from three perspectives: long-term planning, dynamic-immediate planning, and autonomous exploration. These dimensions effectively assess the capabilities of open-world autonomous agents. Researchers in other domains can adopt these perspectives to design comprehensive evaluation tasks for their needs.

---

> ### Author Response · Authors · 2024-11-23
> **Response (2/2)**
>
> **Q1. It would be interesting how well even smaller LMs than Llama 3 8B would perform on Minecraft under the Odyssey framework. Have any experiments of this sort been conducted?**
>
>
> Thanks for your constructive comments. As suggested, we have additionally compared smaller LLMs (LLaMA-3-1B and LLaMA-3-3B) on the single-round long-term planning tasks. The results in Table R1 show that larger model parameters lead to better agent performance. Specifically, LLaMA-3-3B generally performs better than LLaMA-3-1B, achieving higher success rates across the tasks. This additional experiement have been added in the revision (**Page 37, Appendix F.4**).
>
>
>
> Table R1. Performance comparison of smaller LLMs on the single-round long-term planning task. All evaluation metrics are calculated only for successful tasks. $\pm$ corresponds to one standard deviation of the average evaluation over successful tasks.
> | Task                 | Model            | Success Rate | Health     | Time (min)      | # LLM Iters   |
> | -------------------- | ---------------- | ------------ | ---------- | ---------- | ----------- |
> | 1 zombie             | LLaMA-3-3B        | 4 / 8        | 20.0 $\pm$ 0.0 | 19.4 $\pm$ 5.3 | 12.5 $\pm$ 5.1  |
> |                      | LLaMA-3-1B        | 2 / 8        | 20.0 $\pm$ 0.0 | 9.4 $\pm$ 2.7  | 9.5 $\pm$ 4.9   |
> | 1 skeleton           | LLaMA-3-3B        | 4 / 8        | 18.5 $\pm$ 1.5 | 11.5 $\pm$ 7.2 | 12.5 $\pm$ 6.4  |
> |                      | LLaMA-3-1B        | 3 / 8        | 19.6 $\pm$ 0.5 | 14.0 $\pm$ 8.6 | 10.3 $\pm$ 9.2  |
> | 1 spider             | LLaMA-3-3B        | 3 / 8        | 18.1 $\pm$ 3.3 | 9.1 $\pm$ 2.8  | 9.7 $\pm$ 5.7   |
> |                      | LLaMA-3-1B        | 2 / 8        | 19.5 $\pm$ 0.7 | 9.8 $\pm$ 1.3  | 8.5 $\pm$ 6.4   |
>
>
> ---
>
> We hope that these revisions and clarifications address your concerns. Looking forward to your re-evaluation.
>
>
> ---
>
> **References**
>
>
>
>
> [1] Automatic decomposition of planned assembly sequences into skill primitives. IEEE transactions on Robotics and Automation, 2001.
>
> [2] Robot skills for manufacturing: From concept to industrial deployment." Robotics and Computer-Integrated Manufacturing, 2016.
>
> [3] AutoWebGLM: A Large Language Model-based Web Navigating Agent. SIGKDD 2024.
>
> [4] Weblinx: Real-world website navigation with multi-turn dialogue. ICML 2024.
>
> [5] Code as Policies: Language Model Programs for Embodied Control. ICRA, 2023.
>
> [6] ProgPrompt: Generating Situated Robot Task Plans using Large Language Models. ICRA, 2023.
>
> [7] NavGPT: Explicit Reasoning in Vision-and-Language Navigation with Large Language Models. AAAI, 2024.
>
> [8] Navigation with Large Language Models: Semantic Guesswork as a Heuristic for Planning. CoRL, 2023.
>
> [9] A Survey on Game Playing Agents and Large Models: Methods, Applications, and Challenges. arXiv, 2024.
>
> [10] Retrieval-Augmented Generation for Knowledge-Intensive NLP Tasks. NeurIPS, 2020.

---

> > ### Comment · Reviewer_bSai · 2024-11-24
> > **Reviewer Response**
> >
> > I appreciate the authors response, and the new experiments testing Odyssey for smaller LLMs. I will increase my "contribution" score for the paper.

---

> > > ### Author Response · Authors · 2024-11-25
> > > **Thanks for your positive support**
> > >
> > > Thanks for your positive support. Have a nice day :)

---

### Official Review · Reviewer_uziC · 2024-11-04

**Soundness:** 2
**Presentation:** 2
**Contribution:** 1
**Rating:** 3
**Confidence:** 5

**Summary:**

The ODYSSEY framework enhances LLM-based agents in Minecraft by equipping them with an extensive open-world skill library and fine-tuning a LLaMA-3 model using a large Minecraft-specific dataset. It introduces a new benchmark to evaluate agent capabilities in long-term planning, dynamic planning, and autonomous exploration. ODYSSEY outperforms previous methods in adaptability and efficiency, offering a cost-effective solution for open-world agent research.

**Strengths:**

1.	The visual illustrations are appealing and elaborate.
2.	The appendix provides a thorough and detailed explanation of the methods.

**Weaknesses:**

1.	ODYSSEY’s pipeline is highly similar to existing frameworks such as Voyager, Optimus-1[1], and ADAM[2].
2.	ODYSSEY relies on predefined primitive skills, which were generated by GPT-4, whereas GPT-4 itself can directly write JavaScript programs based on Mineflayer. This approach of relying on primitive skills limits the agent’s ability to perform more complex and open-ended tasks, such as building.
3.	On programmatic tasks, ODYSSEY does not demonstrate a broader task range compared to baselines, remaining at the diamond level, already achievable by Voyager. What about more difficult tasks?
4.	The comparisons shown in Table 3 are unfair, as DEPS and VPT use keyboard and mouse as action spaces, rather than JavaScript code, and VPT additionally utilizes visual observation. This is fundamentally different from ODYSSEY, which uses privileged information as its observation space, making such comparisons invalid.
5.	The authors fine-tuned LLaMA-3 on a supplementary dataset (Minecraft Wiki) to create MineMA, but in Tables 4 and 5, the comparison is made against open-source models of equivalent size that lack Minecraft-specific knowledge, resulting in weaker performance. I suggest comparing MineMA with models like GPT and Claude, which possess robust Minecraft knowledge, to demonstrate the significance and efficacy of the additional fine-tuning.
6.	Several related works were not cited, including:
	•	[1] Optimus-1: Hybrid Multimodal Memory Empowered Agents Excel in Long-Horizon Tasks
	•	[2] ADAM: An Embodied Causal Agent in Open-World Environments
	•	[3] OmniJARVIS: Unified Vision-Language-Action Tokenization Enables Open-World Instruction Following Agents, NeurIPS 2024
	•	[4] Steve-Eye: Equipping LLM-based Embodied Agents with Visual Perception in Open Worlds, ICLR 2024

**Questions:**

See the weakness

---

> ### Author Response · Authors · 2024-11-23
> **Response (1/2)**
>
> We greatly appreciate the reviewer for the insightful comments on Odyssey, which helped improve the quality of the paper significantly. We have carefully revised the manuscript according to your valuable suggestions. Below we address the main points raised in the review.
>
> **W1. ODYSSEY’s pipeline is highly similar to existing frameworks such as Voyager, Optimus-1[1], and ADAM[2].**
>
>
> Thanks for your comment. It is worth noting that our focus is not to design a new LLM-based agent architecture. Instead, this work aims to provide a comprehensive framework for developing and evaluating autonomous agents in open-world environments, enabling them to explore the vast and diverse Minecraft world.
>
> (1) Existing efforts mainly focus on solving basic programmatic tasks, like material collection and tool-crafting. This limitation stems from the narrowly defined set of actions available to agents, requiring them to learn effective long-horizon strategies from scratch. Thus, discovering diverse gameplay opportunities in the open world becomes challenging. LLM-based Voyager advanced exploration using skill generation but relied on GPT-4, incurring substantial costs (reaching thousands of dollars per experiment). This expense can be prohibitive for many researchers. In response, we develop a skill library that agents can call upon, achieving performance comparable to Voyager with GPT-4, but using only 8B LLMs. This makes our approach significantly more accessible and cost-effective. Our work facilitates deeper research within the Minecraft environment, extending beyond basic programmatic tasks. We believe this potential for broader exploration represents a major contribution to the research community.
>
> (2) Please note that Optimus-1 and ADAM are our concurrent work (Optimus-1 was published on arXiv on Aug 7, 2024, and updated on Oct 21, 2024; ADAM was published on arXiv on Oct 29, 2024, while the ICLR submission date was Oct 1, 2024).
>
>
> **W2. ODYSSEY relies on predefined primitive skills, which were generated by GPT-4, whereas GPT-4 itself can directly write JavaScript programs based on Mineflayer. This approach of relying on primitive skills limits the agent’s ability to perform more complex and open-ended tasks, such as building.**
>
> Sorry for the misunderstanding.
>
> (1) Our skill library not only includes primitive skills but also features 183 compositional skills that Voyager lacks. Compositional skills encapsulate primitive skills into higher-level tasks, addressing various basic programmatic tasks like mineDiamond and craftIronPickaxe. While Voyager builds its skill library by allowing the LLM to create skills from a limited set of primitive skills, this leads to unstable skill generation and high learning costs. In contrast, Odyssey offers an open-world skill library with both primitive and compositional skills, enabling LLM-based agents to efficiently generate complex policies for broader exploration and more complex tasks.
>
>
> (2) We believe that providing a set of skills is not limited but rather essential for practical open-world scenarios. In real-world applications, it is unrealistic to expect an agent to learn all tasks from scratch. Pre-existing skills allow the agent to efficiently handle simpler tasks, thereby enabling it to focus on more complex challenges. Please note that our skill library also includes basic building-related skills, such as crafting and placing, which the agent can use to complete complex building tasks. It is important to emphasize that the tasks in our benchmark cannot be simply solved by any single skill. Instead, they require a sophisticated combination of multiple skills, ensuring that the benchmark remains challenging.
>
>
> **W3. On programmatic tasks, ODYSSEY does not demonstrate a broader task range compared to baselines, remaining at the diamond level, already achievable by Voyager. What about more difficult tasks?**
>
> Thanks for your comment. Early Minecraft research often focuses on basic programmatic tasks, treating the ObtainDiamond task as the main challenge. This approach only assesses the ability of agents to prioritize crafting steps within a limited task space, rather than their potential for developing complex and diverse solutions. Therefore, this paper does not focus on basic programmatic tasks; we use them merely to simply verify the effectiveness of the proposed skill library. Our benchmark introduces three distinct tasks to evaluate the planning capabilities of agents in Minecraft: long-term planning tasks, which assess the agent's ability to formulate and execute comprehensive plans over extended periods; dynamic-immediate planning tasks, which require agents to adapt their plans dynamically based on immediate environmental feedback; and autonomous exploration tasks, which evaluate the agent's ability to explore the Minecraft world independently, without specific goals.

---

> ### Author Response · Authors · 2024-11-23
>
> **W4. The comparisons shown in Table 3 are unfair, as DEPS and VPT use keyboard and mouse as action spaces, rather than JavaScript code, and VPT additionally utilizes visual observation. This is fundamentally different from ODYSSEY, which uses privileged information as its observation space, making such comparisons invalid.**
>
> Thanks for your valuable feedback. We have removed Table 3 in the revision. In our experiments on the open-world skill library (**Page 6, Section 5.1**), we have updated the experiments to focus solely on testing the effectiveness of directly invoking different skills (without using an agent, and therefore not directly comparing with existing agent methods). The results in Table 2 demonstrate that our open-world skill library efficiently handles basic programmatic task.
>
>
> **W5. The authors fine-tuned LLaMA-3 on a supplementary dataset (Minecraft Wiki) to create MineMA, but in Tables 4 and 5, the comparison is made against open-source models of equivalent size that lack Minecraft-specific knowledge, resulting in weaker performance. I suggest comparing MineMA with models like GPT and Claude, which possess robust Minecraft knowledge, to demonstrate the significance and efficacy of the additional fine-tuning.**
>
> Thanks for your valuable suggestion. We have additionally compared our MineMA with powerful GPT-4o. The results in Table 4 show that our MineMA can achieve performance similar to that of GPT-4o. The corresponding tables and analysis have been updated in the revision (**Page 8--9, Section 5.2.2**).
>
>
>
> Table 4. Performance comparison of different models on the dynamic-immediate planning task.
> | Task                                    | Model         | Success Rate | Time (min)       | # LLM Iters       |
> |-----------------------------------------|---------------|--------------|------------------|-------------------|
> |Collect Seeds|GPT-4o|5/5|1.2±0.5|1.0±0.0|
> ||Baichuan2-7B|2/5|1.8±1.4|3.0±2.8|
> ||Qwen2-7B|2/5|3.8±1.5|4.5±0.7|
> ||MineMA-8B|**5/5**|**1.3±1.4**|*1.4±0.9*|
> ||MineMA-70B|**5/5**|*1.4±1.6*|**1.0±0.0**|
> |Hoe Farmland|GPT-4o|5/5|3.9±3.3|5.8±4.7|
> ||Baichuan2-7B|0/5|N/A|N/A|
> ||Qwen2-7B|*2/5*|*15.7±16.2*|*19.5±10.6*|
> ||MineMA-8B|*2/5*|17.2±14.7|26.5±9.2|
> ||MineMA-70B|**4/5**|**10.2±6.7**|**11.8±2.6**|
> |Shear Sheep|GPT-4o|5/5|4.7±3.6|5.6±6.5|
> ||Baichuan2-7B|1/5|26.0±0.0|30.0±0.0|
> ||Qwen2-7B|*2/5*|*11.0±2.8*|**10.8±1.5**|
> ||MineMA-8B|*2/5*|15.7±10.9|13.0±9.9|
> ||MineMA-70B|**3/5**|**6.9±7.8**|*11.0±7.5*|
> |Milk Cow|GPT-4o|3/5|17.9±8.3|20.3±9.1|
> ||Baichuan2-7B|0/5|N/A|N/A|
> ||Qwen2-7B|*1/5*|26.1±0.0|30.0±0.0|
> ||MineMA-8B|*1/5*|**7.2±0.0**|**7.0±0.0**|
> ||MineMA-70B|**2/5**|*8.6±10.0*|*10.0±11.3*|
> |Cook Meat|GPT-4o|3/5|5.5±2.7|5.0±4.2|
> ||Baichuan2-7B|0/5|N/A|N/A|
> ||Qwen2-7B|0/5|N/A|N/A|
> ||MineMA-8B|*1/5*|*25.6±0.0*|*38.0±0.0*|
> ||MineMA-70B|**2/5**|**20.2±8.5**|**24.0±2.8**|
> |Obtain Leather|GPT-4o|5/5|14.8±10.4|13.0±8.2|
> ||Baichuan2-7B|0/5|N/A|N/A|
> ||Qwen2-7B|1/5|*14.9±0.0*|*16.0±0.0*|
> ||MineMA-8B|*4/5*|15.0±8.7|17.8±15.2|
> ||MineMA-70B|**5/5**|**7.4±7.8**|**8.8±8.6**|
> |Make Sugar|GPT-4o|5/5|5.5±3.6|7.0±2.4|
> ||Baichuan2-7B|2/5|16.2±15.6|22.0±18.4|
> ||Qwen2-7B|2/5|15.4±7.0|15.5±9.2|
> ||MineMA-8B|**5/5**|**4.3±1.9**|**7.0±1.9**|
> ||MineMA-70B|**5/5**|*4.3±4.4*|*7.8±4.0*|
> |Collect Water|GPT-4o|5/5|11.4±1.6|27.3±6.7|
> ||Baichuan2-7B|0/5|N/A|N/A|
> ||Qwen2-7B|1/5|*10.0±0.0*|10.0±0.0|
> ||MineMA-8B|*4/5*|10.4±3.0|**8.8±5.5**|
> ||MineMA-70B|**5/5**|**9.3±4.8**|*9.4±3.7*|
>
>
> **W6. Several related works were not cited, including: [1] Optimus-1: Hybrid Multimodal Memory Empowered Agents Excel in Long-Horizon Tasks [2] ADAM: An Embodied Causal Agent in Open-World Environments [3] OmniJARVIS: Unified Vision-Language-Action Tokenization Enables Open-World Instruction Following Agents, NeurIPS 2024 [4] Steve-Eye: Equipping LLM-based Embodied Agents with Visual Perception in Open Worlds, ICLR 2024**
>
> Sorry for the missing related works. We have cited these works in the revision (**Page 9, Section 6 Related Works**).
>
>
> ---
>
> We hope that these revisions and clarifications address your concerns. Looking forward to your re-evaluation.

---

### Official Review · Reviewer_YBGA · 2024-11-05

**Soundness:** 3
**Presentation:** 3
**Contribution:** 2
**Rating:** 5
**Confidence:** 4

**Summary:**

Manuscript presents several contributions toward building more capable agents in open-world Minecraft: 1) a primitive (and compositional) library of scripted skills; 2) A fine-tuned LLaMA-3 model on QA dataset curated from Minecraft wiki; 3) A new agent benchmark including various tasks in Minecraft. Experiments on programmatic tasks and the tasks in the proposed benchmark show promises over prior arts and counterparts LLMs.

**Strengths:**

+Overall the paper is clearly written, the graphics are stylish and the write-up is good.

+The research topic (open-world agents, LLMs, etc) is relevant to the interest of NeurIPS community.

+The proposed benchmark is interesting and somewhat comprehensive in terms of the diversity and complexity of tasks and the open-world capabilities that can be evaluated.

**Weaknesses:**

-The contributions, though they require a considerable amount of work, do not constitute the significance needed by a conference paper of a top-tier conference like ICLR. Indeed I found the three pillars: the primitive skill library, the LLM for Minecraft QA, and the benchmark are loosely connected and it is unclear how they can benefit better open-world Minecraft agents as a whole.

More importantly, it does not look obvious to me how can these pillars be distinguished from several prior works on similar fronts -- the concept of primitive skills has been introduced by at least a few times including DEPS (Wang et al., 2023), Voyager (Wang et al., 2023), Plan4MC, etc, in both scripted and end-to-end control fashion; the fine-tuned LLM for Minecraft QA can be found in OmniJARVIS (Wang et al., 2024), etc; the benchmark is even more frequently explored in BASALT, MineDoJo, Voyager, DEPS, GROOT (Cai et al., 2023), GROOT-2 (Cai et al., 2024). In the rebuttal, I do expect a comprehensive review of how the contribution presented in the manuscript can be more significant than these for building better open-world agents.

-The results in table 3 should be more carefully examined, as two of the three baselines indeed employ end-to-end control rather than scripted skills. Without an ablation on this, it cannot justify the effectiveness of the proposed method, at least on programmatic tasks.

**Questions:**

See Weaknesses.

---

> ### Author Response · Authors · 2024-11-23
> **Response**
>
> We greatly appreciate the reviewer for the insightful comments on Odyssey, which helped improve the quality of the paper significantly. We have carefully revised the manuscript according to your valuable suggestions. Below we address the main points raised in the review.
>
> **W1. In the rebuttal, I do expect a comprehensive review of how the contribution presented in the manuscript can be more significant than these for building better open-world agents.**
>
> Thanks for your valuable comments. We would like to emphasize that our contribution does not lie in technical novelty. This is why we select "datasets and benchmarks" as the primary area in the OpenReview submission, which is more suited for contributions like ours that offer new datasets, benchmarks, and tools to the research community. The proposed Odyssey consists of three key contributions:
>
>
>
> - **An interactive agent equipped with an open-world skill library.** Existing efforts mainly focus on solving basic programmatic tasks, like material collection and tool-crafting. This limitation stems from the narrowly defined set of actions available to agents, requiring them to learn effective long-horizon strategies from scratch. Thus, discovering diverse gameplay opportunities in the open world becomes challenging. LLM-based Voyager advanced exploration using skill generation but relied on GPT-4, incurring substantial costs (reaching thousands of dollars per experiment). This expense can be prohibitive for many researchers. In response, we develop a skill library that agents can call upon, achieving performance comparable to Voyager with GPT-4, but using only 8B LLMs. This makes our approach significantly more accessible and cost-effective. Our work facilitates deeper research within the Minecraft environment, extending beyond basic programmatic tasks. We believe this potential for broader exploration represents a major contribution to the research community.
>
> - **A fine-tuned LLaMA-3 model trained on a large-scale question-answering dataset.** Our dataset involves 390k+ instruction entries derived from the Minecraft Wiki. In contrast, the Wiki dataset released by MineDojo only collects Minecraft Wiki pages, without refining the content and generating Q&A pairs for model training. STEVE introduces a non-public dataset with 20k+ Q&A pairs, which is smaller than our dataset in terms of scale and diversity. The OmniJARVIS dataset is currently not public, and without knowing its exact size, we cannot make a comparison. It is notable that OmniJARVIS is our concurrent work (it was published on arXiv on Jun 27, 2024, and updated on Oct 31, 2024, while the ICLR submission date was Oct 1, 2024).
>
> - **A new agent capability benchmark encompassing a variety of tasks.** Existing benchmark with programmatic tasks only assess the ability of agents to prioritize crafting steps within a limited task space, rather than their potential for developing complex and diverse solutions. Our benchmark introduces three distinct tasks to evaluate the planning capabilities of agents in Minecraft: long-term planning tasks, which assess the agent ability to formulate and execute comprehensive plans over extended periods; dynamic-immediate planning tasks, which require agents to adapt their plans dynamically based on immediate environmental feedback; and autonomous exploration tasks, which evaluate the agent ability to explore the Minecraft world independently, without specific goals. It is notable that GROOT-2 is also our concurrent work, which is under anonymous review in ICLR 2025.
>
>
>
>
> **W2. The results in table 3 should be more carefully examined, as two of the three baselines indeed employ end-to-end control rather than scripted skills. Without an ablation on this, it cannot justify the effectiveness of the proposed method, at least on programmatic tasks.**
>
> Thanks for your valuable feedback. We have removed Table 3 in the revision. In our experiments on the open-world skill library (**Page 6, Section 5.1**), we have updated the experiments to focus solely on testing the effectiveness of directly invoking different skills (without using an agent, and therefore not directly comparing with existing agent methods). The results in Table 2 demonstrate that our open-world skill library efficiently handles basic programmatic task. Simple tasks achieve near-perfect success within five minutes. Even for difficult tasks like obtaining a diamond, success rates rise from 21.7\% at five minutes to 92.5\% at ten minutes, highlighting the effectiveness of the skill library.
>
> ---
>
> We hope that these revisions and clarifications address your concerns. Looking forward to your re-evaluation.

---

> > ### Author Response · Authors · 2024-11-29
> > **Looking forward to your reevaluation**
> >
> > Dear Reviewer YBGA,
> >
> > We are glad that the reviewer appreciates our attempt, and sincerely thank the reviewer for the constructive comments. As suggested, we have additionally added some clarifications about our contributions and updated our experiments on the open-world skill library. Please let us know if you have other questions or comments.
> >
> > We sincerely look forward to your reevaluation of our work and would very appreciate it if you could raise your score to boost our chance of more exposure to the community. Thank you very much!
> >
> > Best regards,
> >
> > The authors of Odyssey

---

### Official Review · Reviewer_REj7 · 2024-11-05

**Soundness:** 2
**Presentation:** 2
**Contribution:** 2
**Rating:** 3
**Confidence:** 3

**Summary:**

This paper addresses the development and evaluation of generalist agents in open-world environments like Minecraft. The authors introduce Odyssey, a framework that equips LLM-based agents with enhanced open-world skills to enable more diverse exploration. Odyssey includes (1) an agent skill library with 40 primitive and 183 compositional skills, (2) a fine-tuned LLaMA-3 model trained on Minecraft Wiki instructions, and (3) a new benchmark covering long-term planning, dynamic planning, and autonomous exploration tasks. Experiments show Odyssey’s effectiveness in evaluating agent capabilities. All resources are publicly available to support future research on autonomous agents.

**Strengths:**

1. This paper demonstrates substantial effort, including the collection of Minecraft-specific data, fine-tuning a large language model, building a Minecraft agent, comparing it with numerous baselines, and designing three evaluation benchmarks.
2. The paper is well-formatted, with clear and coherent expression of ideas, making it easy for readers to follow and understand.

**Weaknesses:**

I strongly agree with the paper’s critique that “current research in Minecraft is overly focused on tasks like mining diamonds.” Minecraft is indeed a valuable platform for studying generalist agents, as it simulates numerous real-world challenges such as complex perception, an infinite task space, partial observability, and intricate terrains—all unsolved issues. Developing agents in Minecraft should ideally contribute towards generalization in other environments, even the real world. However, much of the current research overlooks these challenges, using scripted, privilege-enabled setups like Mineflayer to turn Minecraft into a text-based game. This approach often revolves around how to prompt large language models like GPT-4 to decompose long-horizon tasks, which isn’t easily transferable to other settings, as few environments provide global privileged information or powerful controllers like Mineflayer. Although there are numerous studies of this kind, they rarely yield new insights, and unfortunately, this paper falls into this paradigm.

1. The paper repeatedly emphasizes that “our focus is not to design a new LLM-based agent architecture.” However, a significant portion is still dedicated to detailing the agent architecture, even listing it as part of the contribution. Since this architecture is not novel, it would be better suited to the appendix.
2. Given that the focus is not on a “new LLM-based agent architecture,” performing an ablation study on a standard architecture seems less meaningful.
3. The comparison in Table 3 is inherently unfair. The VPT model operates in the native, unmodified environment with RGB output and mouse and keyboard controls, while GITM and the proposed work use Mineflayer as a controller.
4. Fine-tuning on Minecraft-specific knowledge is expected to improve performance compared to large, untuned models, so this result is unsurprising.

**Questions:**

Refer to the weakness.

---

> ### Author Response · Authors · 2024-11-23
> **Response**
>
> We greatly appreciate the reviewer for the insightful comments on Odyssey, which helped improve the quality of the paper significantly. We have carefully revised the manuscript according to your valuable suggestions. Below we address the main points raised in the review.
>
> ---
>
> **W1. The paper repeatedly emphasizes that “our focus is not to design a new LLM-based agent architecture.” However, a significant portion is still dedicated to detailing the agent architecture, even listing it as part of the contribution. Since this architecture is not novel, it would be better suited to the appendix.**
>
> Sorry for the confusion. As stated in the Introduction (**Page 2, Line 105--106**), this work aims to provide a comprehensive framework for developing and evaluating autonomous agents in open-world environments. We believe that a brief explanation of the agent architecture is crucial for understanding how our framework integrates with the existing architecture to achieve the results we present. To address your concern, we have condensed the section on the agent architecture to approximately half a page in the revision (**Page 4, Section 2.2**).
>
> **W2. Given that the focus is not on a "new LLM-based agent architecture," performing an ablation study on a standard architecture seems less meaningful.**
>
> Sorry for the confusion. As mentioned in our response to W1, one of our main contributions is the evaluation of autonomous agents. Our LLM-based agent employs a planner-actor-critic architecture to facilitate task decomposition, skill execution, and performance feedback. This is a widely used general framework, and since our goal is to assess the agent performance on the proposed benchmark, it is essential to test the effectiveness and robustness of the different components within the agent. This provides a necessary reference point for understanding how each part contributes to the overall performance.
>
>
> **W3. The comparison in Table 3 is inherently unfair. The VPT model operates in the native, unmodified environment with RGB output and mouse and keyboard controls, while GITM and the proposed work use Mineflayer as a controller.**
>
> Thanks for your valuable feedback. We have removed Table 3 in the revision. In our experiments on the open-world skill library (**Page 6, Section 5.1**), we have updated the experiments to focus solely on testing the effectiveness of directly invoking different skills (without using an agent, and therefore not directly comparing with existing agent methods). The results in Table 2 demonstrate that our open-world skill library efficiently handles basic programmatic task.
>
>
> **W4. Fine-tuning on Minecraft-specific knowledge is expected to improve performance compared to large, untuned models, so this result is unsurprising.**
>
> Thanks for your valuable comments. We would like to emphasize that our contribution does not lie in technical novelty. This is why we select "datasets and benchmarks" as the primary area in the OpenReview submission, which is more suited for contributions like ours that offer new datasets, benchmarks, and tools to the research community. Therefore, one of our main contributions is providing the fine-tuned LLaMA-3 model and a comprehensive question-answering dataset. The experiments are conducted merely to validate the effectiveness of the model and data we provide.
>
> Existing researchers can utilize our fine-tuned LLaMA-3 model to construct their own Minecraft agents. Additionally, they can leverage our dataset to further fine-tune models tailored to their specific requirements. We have also open-sourced our web crawler program on GitHub, which was used to collect data from Wikis. Researchers can modify this program to crawl data relevant to their needs.
>
> ---
>
> We hope that these revisions and clarifications address your concerns. Looking forward to your re-evaluation.

---

> > ### Comment · Reviewer_REj7 · 2024-11-25
> >
> > I appreciate the authors’ responses and their consideration of my review suggestions. However, I am not particularly interested in the approach of using large language models to drive MineFlayer and turn Minecraft into a text-based game. Therefore, I will maintain my initial evaluation.

---

> > > ### Author Response · Authors · 2024-11-29
> > > **Looking forward to your reevaluation**
> > >
> > > Dear Reviewer REj7,
> > >
> > > We are truly honored by the reviewer's kind acknowledgment of our efforts, and we sincerely express our gratitude for the insightful and constructive comments provided. In line with your valuable latest suggestions, we have sincerely provided further clarifications regarding the use of MineFlayer with text-based game. We would be deeply grateful if you could kindly let us know if there are any further questions or comments.
> > >
> > > Best regards,
> > >
> > > The authors of Odyssey

---

> ### Author Response · Authors · 2024-11-25
> **Thanks for your feedback.**
>
> We are deeply honored to receive your response and your recognition of our work, and we sincerely appreciate your thoughtful feedback.
>
> (1) We consider the choice of whether to use MineFlayer with text-based game as two different paradigm for building a Minecraft agent, essentially differing in how the observation space and action space is defined. We believe that providing a set of skills is not a limitation but rather essential for practical open-world scenarios. In real-world applications, it is unrealistic to expect an agent to learn all tasks from scratch. Pre-existing skills allow the agent to efficiently handle simpler tasks, thereby enabling it to focus on more complex challenges. We believe that one of our significant contributions is indeed the engineering domain-specific skill library, which represents a meaningful addition to the Minecraft agent community. Developing this library requires significant time and effort in design and debugging, making it a valuable resource for researchers. By leveraging our skill library, even open-source LLMs can enable open-world exploration within the Minecraft environment.
>
> > LLM-based Voyager advances exploration using skill generation but relies on GPT-4, incurring substantial costs (reaching thousands of dollars per experiment). This expense can be prohibitive for many researchers. On the contrary, our agent can call upon the proposed skill library, achieving performance comparable to Voyager with GPT-4, but using only 8B LLMs.
>
> (2) We would like to emphasize that our contribution does not lie in technical novelty. This is why we select "datasets and benchmarks" as the primary area in the OpenReview submission, which is more suited for contributions like ours that offer new datasets, benchmarks, and tools to the research community. The proposed Odyssey consists of three key contributions: (a) a fine-tuned LLaMA-3 model trained on a large-scale question-answering dataset; (b) an interactive agent equipped with an open-world skill library; (c) a new agent capability benchmark encompassing a variety of tasks. We have open-sourced all parts of ODYSSEY (see supplementary material) and will continuously update the repository. While our work may not directly align with your interests, we hope this will enable other researchers to build upon our work, fostering further innovation and progress in the development of autonomous agents.
>
> We sincerely look forward to your reevaluation of our work and would very appreciate it if you could raise your score to boost our chance of more exposure to the community. Thank you once again :)

---

### Comment · Area_Chair_Nd4F · 2024-11-25

Dear Reviewers,


This is a friendly reminder that the discussion will end on Nov. 26th (anywhere on Earth). If you have not already, please take a close look at all reviews and author responses, and comment on whether your original rating stands.


Thanks,

AC

---

### Note · Authors · 2025-01-22

I have read and agree with the venue's withdrawal policy on behalf of myself and my co-authors.